# BRCA2 BRC missense variants disrupt RAD51-dependent DNA repair

Judit Jimenez-Sainz[1], Joshua Mathew[1], Gemma Moore[1], Sudipta Lahiri[1], Jennifer Garbarino[1], Joseph P Eder[2], Eli Rothenberg[3], Ryan B Jensen[1]*

[1]Department of Therapeutic Radiology, Yale University, New Haven, United States; [2]Department of Medical Oncology, Yale University School of Medicine, Yale Cancer Center, New Haven, United States; [3]Department of Biochemistry and Molecular Pharmacology, New York University, New York, United States

**Abstract** Pathogenic mutations in the BRCA2 tumor suppressor gene predispose to breast, ovarian, pancreatic, prostate, and other cancers. BRCA2 maintains genome stability through homology-directed repair (HDR) of DNA double-strand breaks (DSBs) and replication fork protection. Nonsense or frameshift mutations leading to truncation of the BRCA2 protein are typically considered pathogenic; however, missense mutations resulting in single amino acid substitutions can be challenging to functionally interpret. The majority of missense mutations in BRCA2 have been classified as Variants of Uncertain Significance (VUS) with unknown functional consequences. In this study, we identified three BRCA2 VUS located within the BRC repeat region to determine their impact on canonical HDR and fork protection functions. We provide evidence that S1221P and T1980I, which map to conserved residues in the BRC2 and BRC7 repeats, compromise the cellular response to chemotherapeutics and ionizing radiation, and display deficits in fork protection. We further demonstrate biochemically that S1221P and T1980I disrupt RAD51 binding and diminish the ability of BRCA2 to stabilize RAD51-ssDNA complexes. The third variant, T1346I, located within the spacer region between BRC2 and BRC3 repeats, is fully functional. We conclude that T1346I is a benign allele, whereas S1221P and T1980I are hypomorphic disrupting the ability of BRCA2 to fully engage and stabilize RAD51 nucleoprotein filaments. Our results underscore the importance of correctly classifying BRCA2 VUS as pathogenic variants can impact both future cancer risk and guide therapy selection during cancer treatment.

*For correspondence: ryan.jensen@yale.edu

Competing interest: The authors declare that no competing interests exist.

## Editor's evaluation

This study provides a thorough functional analysis of three mutations in the BRCA2 gene that do not seem to necessarily cause breast cancer. The authors use functional assays in cancer cells and with recombinant proteins to determine that two BRCA2 variants, S1221P and T1980I, are indeed pathogenic, while the T1346I variant is fully functional and benign. The strength of the study is the rigorous assessment of these mutations in a variety of established assays for BRCA2, and has improved significantly following the review process. The work should have a broad impact in the breast cancer field.

## Introduction

BRCA2 (BReast CAncer type 2 susceptibility gene) is a mediator protein promoting homology-directed repair (HDR) of DNA double-strand breaks (DSBs) via RAD51-dependent DNA strand invasion, pairing, and exchange (*Jensen et al., 2010*; *Scully and Livingston, 2000*; *Venkitaraman, 2002*). BRCA2 binds and loads RAD51 onto single-stranded DNA (ssDNA) resulting from strand resection.

Mechanistically, BRCA2 directs RAD51 onto ssDNA while limiting binding to double-stranded DNA (dsDNA), downregulates the ATPase activity of RAD51, and displaces Replication Protein A (RPA) from ssDNA (*Jensen et al., 2010*; *Liu et al., 2010*; *Thorslund et al., 2007*). Loss of BRCA2 in human cells leads to HDR defects, genomic instability, micronuclei formation, diminished fork protection following replication stress, impaired RAD51 foci formation in response to DNA damage, and sensitivity to chemotherapeutics such as platinum agents and PARP inhibitors (PARPi) (*Abul-Husn et al., 2019*; *Chen et al., 1999*; *Chen et al., 1998*; *Moynahan et al., 2001*; *Patel et al., 1998*).

To date, four recognized domains in BRCA2 have emerged from both sequence based and structural studies (*Figure 1A*): an N-terminal region, eight BRC repeats located in the middle of the protein, an alpha helices region and three tandem oligonucleotide/oligosaccharide-binding folds (OB-folds) termed the DNA binding domain (DBD), and the C-terminal domain (CTD) (*Bignell et al., 1997*; *McAllister et al., 1997*; *Yang et al., 2002*). The BRC repeats interact with RAD51 and distinct functions have been ascribed to each of two modules: BRC1-4 and BRC5-8 (*Carreira and Kowalczykowski, 2011*; *Chatterjee et al., 2016*). The BRC1-4 module binds free monomeric RAD51 while BRC5-8 binds RAD51 only when assembled onto ssDNA (*Carreira and Kowalczykowski, 2011*; *Chatterjee et al., 2016*). The DBD of BRCA2 has been shown to bind DSS1, ssDNA, and presumably dsDNA as well (*Yang et al., 2002*). A nuclear export sequence (NES) in the DBD has been found to overlap with the binding region for DSS1 and a specific missense mutation (D2723H) that disrupts DSS1 binding results in BRCA2 export to the cytoplasm (*Jeyasekharan et al., 2013*). The CTD (also known as TR2) binds RAD51 only when complexed with ssDNA as a nucleoprotein filament (similar to BRC5-8) and RAD51 filament binding is regulated by phosphorylation at the S3291 residue (*Carreira et al., 2009*; *Carreira and Kowalczykowski, 2011*; *Chatterjee et al., 2016*; *Chen et al., 1998*; *Esashi et al., 2007*; *Mizuta et al., 1997*; *Pellegrini et al., 2002*; *Sharan et al., 1997*; *Wong et al., 1997*). Putative nuclear localization signals (NLSs) are located in the CTD of BRCA2; however, surprisingly, it remains unclear how exactly nuclear/cytoplasmic trafficking of BRCA2 is regulated (*Bertwistle et al., 1997*; *Han et al., 2008*; *Spain et al., 1999*; *Yano et al., 2000*) (reviewed in *Jimenez-Sainz and Jensen, 2021*).

The etiology of cancer-causing mutations in BRCA2 is complex with both germline and somatic mutations spanning the entire length of the protein. Nonsense mutations in BRCA2 leading to protein truncation generally predict pathogenicity, as deletion of the DBD would impair key HDR functions, and perhaps more importantly, deletion of NLSs at the C-terminus of BRCA2 result in mislocalization to the cytoplasm (*Spain et al., 1999*). We and others have recently discovered that missense mutations in the DBD mislocalized BRCA2 to the cytosol leading to HR deficiency and sensitivity to crosslinking agents and PARPi (*Jeyasekharan et al., 2013*; *Jimenez-Sainz and Jensen, 2021*; *Jimenez-Sainz et al., 2022*; *Lee et al., 2021*). However, an estimated 80% of germline and somatic mutations identified in patients are missense, single amino acid substitutions in the full-length protein (data extracted from ClinVar and cBioPortal, accessed April 2021) (*Jimenez-Sainz and Jensen, 2021*). Missense mutations often yield unknown impact on function and/or disease linkage and are referred to as Variants of Uncertain Significance (VUS). VUS are frequently found to be unique to individual families, and thus, in the absence of any functional insight, make clinical management of patients extremely challenging. In a recent search of ClinVar (accessed April 2021), more than 60% of the total identified BRCA2 mutations were VUS (*Jimenez-Sainz and Jensen, 2021*). Notably, the majority of the 1388 missense mutations reported within the BRC repeats of BRCA2 are classified as VUS.

The significant role of the BRC repeats in binding, loading, and stabilizing RAD51 on ssDNA prompted us to study the functionality of variants in this domain. Moreover, the biochemical impact of missense mutations in a single BRC repeat, or between repeats, within the context of the full-length BRCA2 protein was unknown. We focused on two variants, S1221P and T1980I, situated within a conserved RAD51 binding core motif in BRC2 and BRC7 respectively. A third variant, T1346I, located in the spacer region between BRC2 and BRC3 was identified as a somatic tumor mutation in a patient undergoing treatment at the Yale Cancer Center (*Lo et al., 2003*; *Olopade et al., 2003*; *Tal et al., 2009*). The variants were previously deposited in ClinVar lacking clinical interpretation and classified as VUS (https://www.ncbi.nlm.nih.gov/clinvar/variation/51501/, https://www.ncbi.nlm.nih.gov/clinvar/variation/455896/, https://www.ncbi.nlm.nih.gov/clinvar/variation/993173/). Our findings reveal S1221P and T1980I substantially alter BRCA2 HDR protein activities that fail to be compensated for by the remaining seven BRC repeats. Altered functions include impaired cellular response to chemotherapeutics and reduced RAD51 foci following ionizing radiation induced DNA damage. We provide

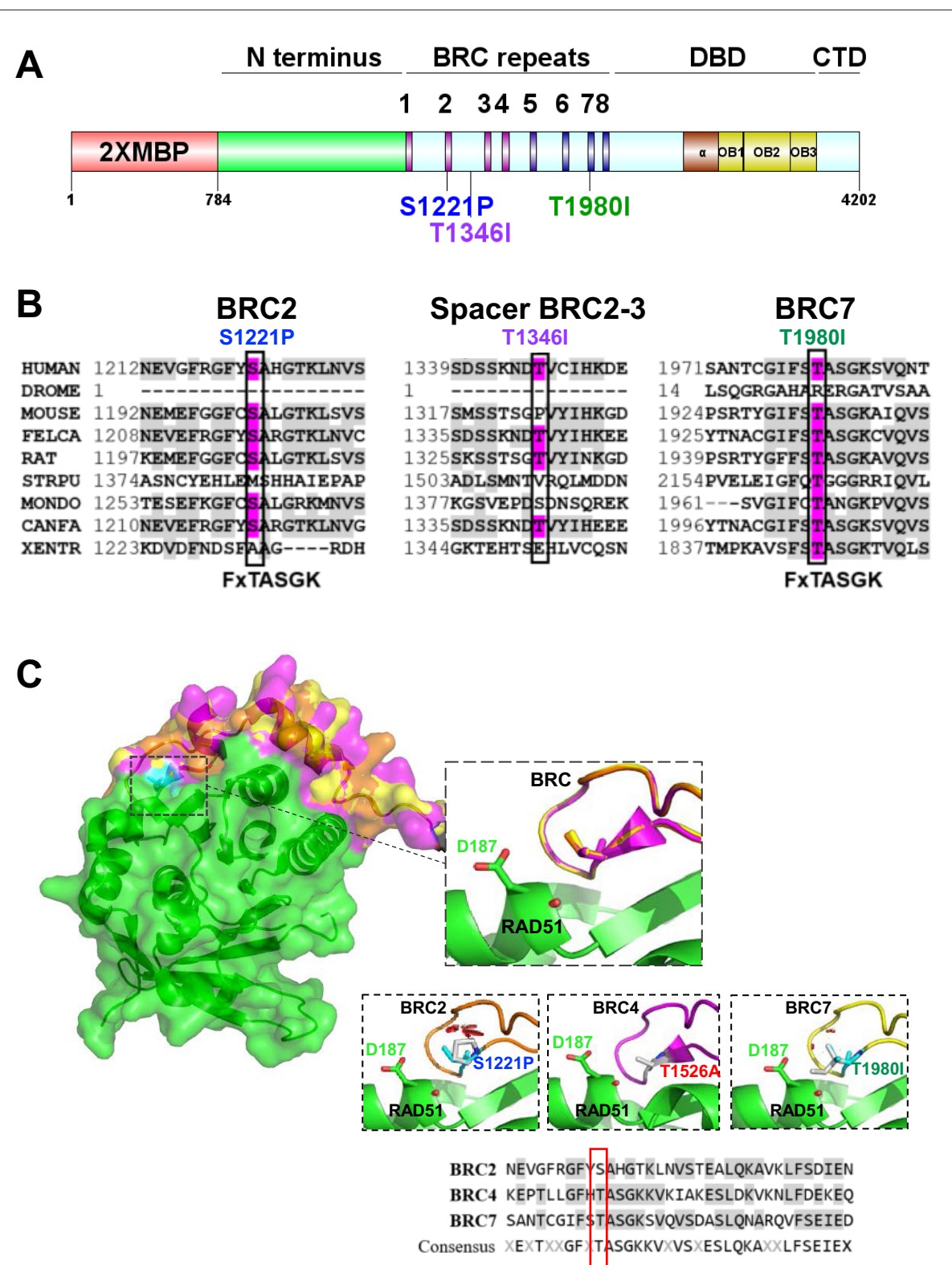

**Figure 1.** BRCA2 BRC residues S1221, T1346, and T1980 are conserved residues. S1221P and T1980I are structurally predicted to disrupt BRC folding and RAD51 binding. (**A**) BRCA2 protein schematic depicting domain organization: 2XMBP tag, N-terminus, BRC repeats, DNA binding domain (DBD), and C-terminal domain (CTD). BRCA2 missense variants used in this study are indicated. (**B**) Multiple sequence alignment of BRCA2 amino acids flanking each missense variant from different organisms: S1221P (BRC2), T1346I (Spacer BRC2-3), and T1980I (BRC7). Residue corresponding to the missense variant is indicated in pink and conserved residues flanking the missense residue are indicated in light grey. Uniprot and ClustalX (70%

*Figure 1 continued on next page*

*Figure 1 continued*

threshold for shading) were used for the alignment. FxTASGK is a consensus motif important for RAD51 interactions depicted below BRC2 and BRC7. (**C**) Structural models based on 1N0W structure. Homology model comparison of BRC2 (orange) and BRC7 (yellow) based on the BRC4 (pink) structure. Top Inset: BRC2 (S1221), BRC4 (T1526), and BRC7 (T1980) overlayed to show conservation. Bottom Insets: predicted disruptions resulting from variants BRC2 (S1221P), BRC4 (T1526A), and BRC7 (T1980I). BRC4 T1526A is a previously characterized missense variant that disrupts RAD51 binding. RAD51 D187 residue is displayed in green. Clashes are indicated in red. Sequence alignment of BRC2, BRC4, and BRC7 repeats. Missense variant residues are boxed in red.

The online version of this article includes the following source data and figure supplement(s) for figure 1:

**Source data 1.** Contains BRCA2 sequence alignment depicting homology (*Figure 1B*), SWISS-MODEL homology modeling reports (*Figure 1C*), close up views of BRC-RAD51 amino acid interactions (*Figure 1C*).

**Figure supplement 1.** Polar contacts of S1221P, T1526A, and T1980I.

**Figure supplement 1—source data 1.** T1526 polar contacts.

unique biochemical evidence that BRC2 S1221P and BRC7 T1980I fail to bind RAD51, disrupt the stabilization of RAD51-ssDNA complexes, and as a result, stimulation of RAD51-mediated DNA strand exchange activity is diminished. In contrast, the T1346I variant resembled the wild type BRCA2 in all respects and was presumably a benign passenger mutation.

## Results

### BRCA2 missense variants S1221P and T1980I predict disruption of BRC folding and RAD51 binding

Searching the ClinVar database, we identified two potentially pathogenic BRCA2 missense variants, S1221P and T1980I, located in BRC2 and BRC7, respectively (*Figure 1A*), and classified as VUS or 'not yet reviewed'. S1221P and T1980I have been described as tumor-associated but detailed information regarding their biochemical functionality was unknown (*Lo et al., 2003*; *Tal et al., 2009*). In addition, we identified a BRCA2 T1346I variant as a somatic mutation from whole exome sequencing (WES) of a colorectal tumor specimen from the Yale Cancer Center. Interestingly, T1346I is located in the spacer region flanked by BRC2 and BRC3 and classified as a VUS. Sequence alignment across 9 different species reveal that the S1221 and T1980 residues are highly conserved (67% and 89% respectively) (*Figure 1B*), whereas T1346 is less well conserved (44%). Multiple studies have verified a conserved core motif within each BRC repeat, designated FxTASGK, that is crucial for RAD51 binding (*Bignell et al., 1997*; *Chen et al., 1998*; *Lo et al., 2003*; *Pellegrini et al., 2002*). Importantly, both the S1221 and T1980 residues are located directly within this motif (note boxed residues in *Figure 1B*). Given that structural information is only available for the BRC4 repeat crystallized with the core domain of RAD51 (*Pellegrini et al., 2002*), we utilized the SWISS-MODEL server homology modelling pipeline which relies on ProMod3, an in-house comparative modelling engine based on OpenStructure (*Biasini et al., 2013*; *Studer et al., 2021*; *Waterhouse et al., 2018*). We based our modeling on the BRC4 structure to determine if S1221P and T1980I would disrupt key contacts between the BRC2 and BRC7 repeats and RAD51 (*Figure 1C*).

Residue T1526 in BRC4 (3rd amino acid position in FxTSAGK) is equivalent to S1221 and T1980 in BRC2 and BRC7, respectively (*Figure 1C*, **lower sequence alignment**). T1526 is buried in a hydrophobic pocket formed by RAD51 (*Figure 1C*, **center panel**) (*Pellegrini et al., 2002*) and the missense mutation, T1526A, diminishes BRC4 functions (*Carreira et al., 2009*). Modeling of the T1526A substitution does not lead to steric clashing, however, could disassemble the BRC4 loop conformation (hydrophobic contacts F1524, A1527 and K1530 and polar contacts T1526 and S1528) necessary for RAD51 binding (*Figure 1—figure supplement 1*; *Carreira et al., 2009*). Mutating S1221 to proline in BRC2 (S1221P) predicts increased steric hindrance resulting in a loop to improperly fold to accommodate the bulkier sidechain (*Figure 1C*, **left panel**). Changing T1980 in BRC7 to isoleucine (T1980I) requires a larger sidechain to be incorporated into the structure but with more flexibility than proline. If T1980I is oriented to not misfold, the sidechain is predicted to block RAD51 binding as it would clash with RAD51 residue D187 (*Figure 1C*, **right panel**). Additionally, S1221P and T1980I substitutions would predict loss of multiple polar contacts (Figure S1) potentially disrupting the BRC structure and RAD51 interactions. Overall, the structural modeling of S1221P and T1980I predicts that either

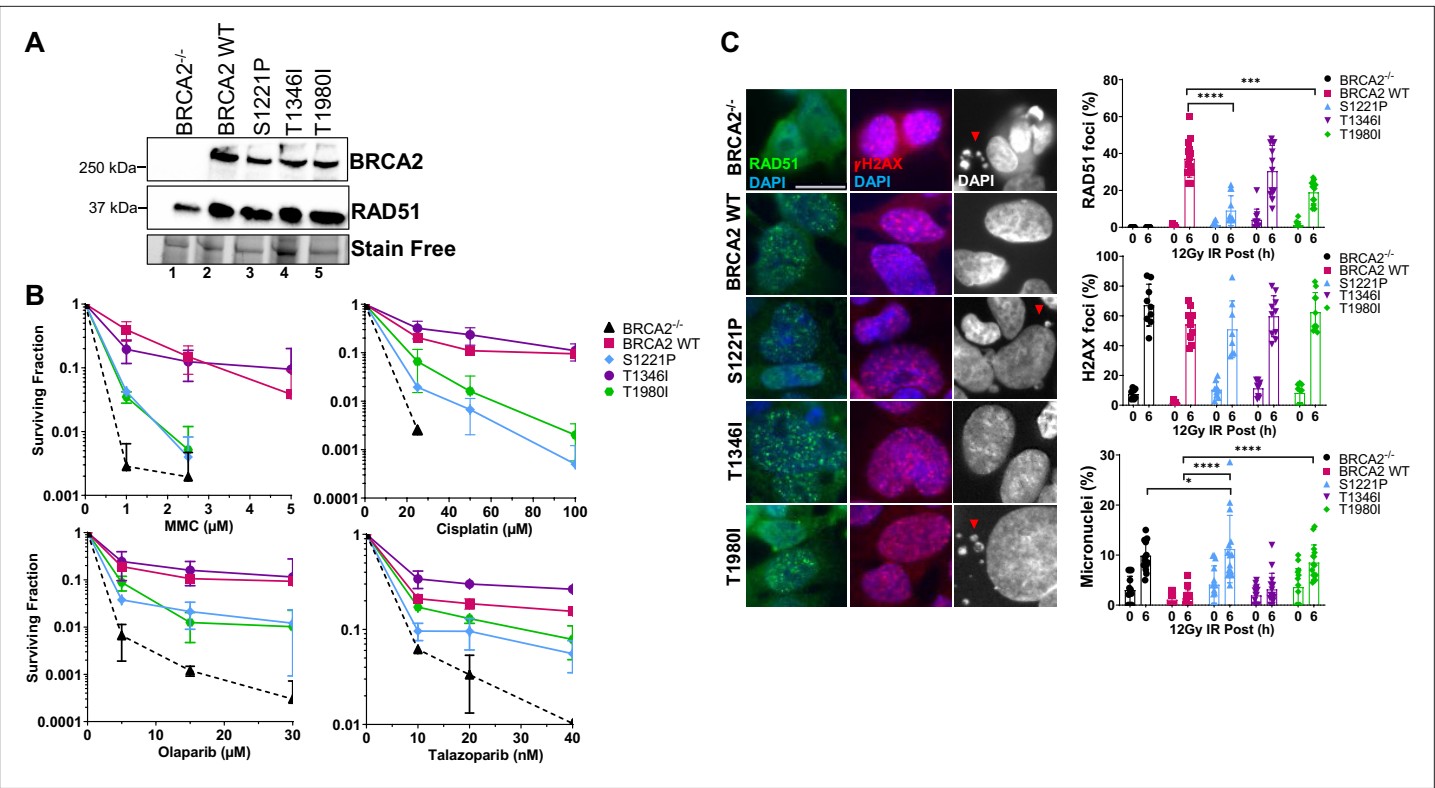

**Figure 2.** S1221P and T1980I only partially complement BRCA2 knockout cells in response to DNA damage. T1346I fully rescues DNA damage sensitivity and RAD51 foci formation upon irradiation. (**A**) Western blot of total cellular lysates from DLD-1 BRCA2$^{-/-}$ cells stably transfected with either empty vector (BRCA2$^{-/-}$) or BRCA2 Wild Type (WT), BRCA2 S1221P (BRC2), T1346I (Spacer BRC2-3), and T1980I (BRC7) full-length BRCA2 cDNA constructs. BRCA2 was detected with an MBP antibody. 2XMBP-BRCA2 (470 kDa), and RAD51 (37 kDa). (**B**) Clonogenic survival analyses of stable cell lines treated with mitomycin C (MMC), cisplatin, Olaparib, and Talazoparib. Error bars represent the S.D. for two biological independent experiments. (**C**) Immunofluorescence images and quantification of RAD51 (green) and gammaH2AX (red) foci, and DAPI staining to visualize nuclei (blue, grey) and micronuclei (grey). Representative images at 6 hr post-IR (12 Gy). Quantification of four biological independent experiments and statistical analysis t-test and one-way ANOVA. Scale bar represents 50 µm. p-value <0.05 p-value <0.0001.

The online version of this article includes the following source data and figure supplement(s) for figure 2:

**Source data 1.** Contains original gel images depicting cropped regions used for *Figure 2A*, clonogenic survival data for *Figure 2B*, foci data for *Figure 2C*.

**Figure supplement 1.** Expression and nuclear localization of S1221P, T1346I, and T1980I BRCA2 proteins.

**Figure supplement 1—source data 1.** Raw Nt Ct BRCA2 RAD51.

**Figure supplement 2.** BRCA2$^{-/-}$ cells stably expressing full-length BRCA2 S1221P and T1980I proteins only partially rescue sensitivity to crosslinking agents (MMC or cisplatin) and PARP inhibitors (Olaparib or Talazoparib).

**Figure supplement 2—source data 1.** Revised plating efficiency.xlsx.

misfolding or steric clashes between residues within the BRC-RAD51 interface will obstruct interactions. As no structural information exists for the spacer region between BRC2 and BRC3 to date, we were unable to model the T1346I mutation.

## S1221P and T1980I variants are defective in response to chemotherapeutics and ionizing radiation

The sequence conservation and homology modeling analysis suggested S1221P and T1980I disrupt RAD51 interactions, therefore, we directly tested whether the variants were capable of functionally complementing BRCA2-deficient cells. Full-length BRCA2 cDNA constructs were fused to a tandem repeat of the maltose binding protein tag (2XMBP) which stabilizes and increases BRCA2 expression as previously described (*Jensen et al., 2010*). We stably expressed the variants in DLD1 BRCA2$^{-/-}$ cells (*Hucl et al., 2008*) and successfully derived single-cell clones expressing S1221P, T1346I, and T1980I

(*Figure 2A*, *Figure 2—figure supplement 1A*). All variants localized to the nucleus (*Figure 2—figure supplement 1B*, **left panel**) as expected. Interestingly, as first described by Baker (*Magwood et al., 2013*) and confirmed in a subsequent paper from our group (*Chatterjee et al., 2016*), re-introduction of recombinant BRCA2 protein in deficient cells increases total expression levels of RAD51 (*Figure 2A*). The basal localization pattern of RAD51 also changed from a diffuse cytoplasmic/nuclear staining in null cells to a bright distinct signal in the nucleus upon re-expression of wild type (WT) BRCA2 and the variants (*Figure 2—figure supplement 1B*, **middle panel**).

Strikingly, in BRCA2 null cells, neither expression of the S1221P nor the T1980I variant was able to rescue survival to the same level as WT BRCA2 in response to crosslinking agents mitomycin C (MMC) and cisplatin (*Figure 2B*). Notably, the two variants did not track with the empty vector control cells (BRCA2$^{-/-}$) suggesting partial complementation. The response of S1221P and T1980I to PARPi (olaparib and talazoparib) was modestly more robust than to crosslinkers but did not rescue to the same extent as WT BRCA2. T1346I provided a full rescue in response to all chemotherapeutic agents tested suggesting T1346I is a benign variant (*Figure 2B*). The differential responses of S1221P and T1980I could not be explained by a difference in plating efficiencies (*Figure 2—figure supplement 2*). Overall, the results imply that S1221P and T1980I are hypomorphic alleles of BRCA2.

RAD51 foci formation in the nuclei of ionizing radiation damaged cells has long been an established biomarker to indicate protein recruitment and functional engagement of homologous recombination at sites of DNA DSBs (*Haaf et al., 1995*; *Rothkamm et al., 2015*). We examined RAD51 foci at 6 hr (peak foci) following treatment with 12 Gy of ionizing radiation (*Figure 2C*). Phosphorylation of gammaH2AX was used as a surrogate marker for DNA DSBs to ensure damage inflicted was comparable amongst all cell lines (*Chatterjee et al., 2016*). WT BRCA2 and T1346I restored RAD51 foci in DLD1 BRCA2 deficient cells whereas S1221P and T1980I exhibited a significantly lower percentage of cells with foci indicating a partial defect in HDR. Micronuclei formation is an indicator of genomic instability and BRCA2 deficient cells, including DLD1 BRCA2 knockout cells, show elevated levels of micronuclei (*Ban et al., 2001*; *Heddle et al., 1983*). While WT BRCA2 and T1346I were able to

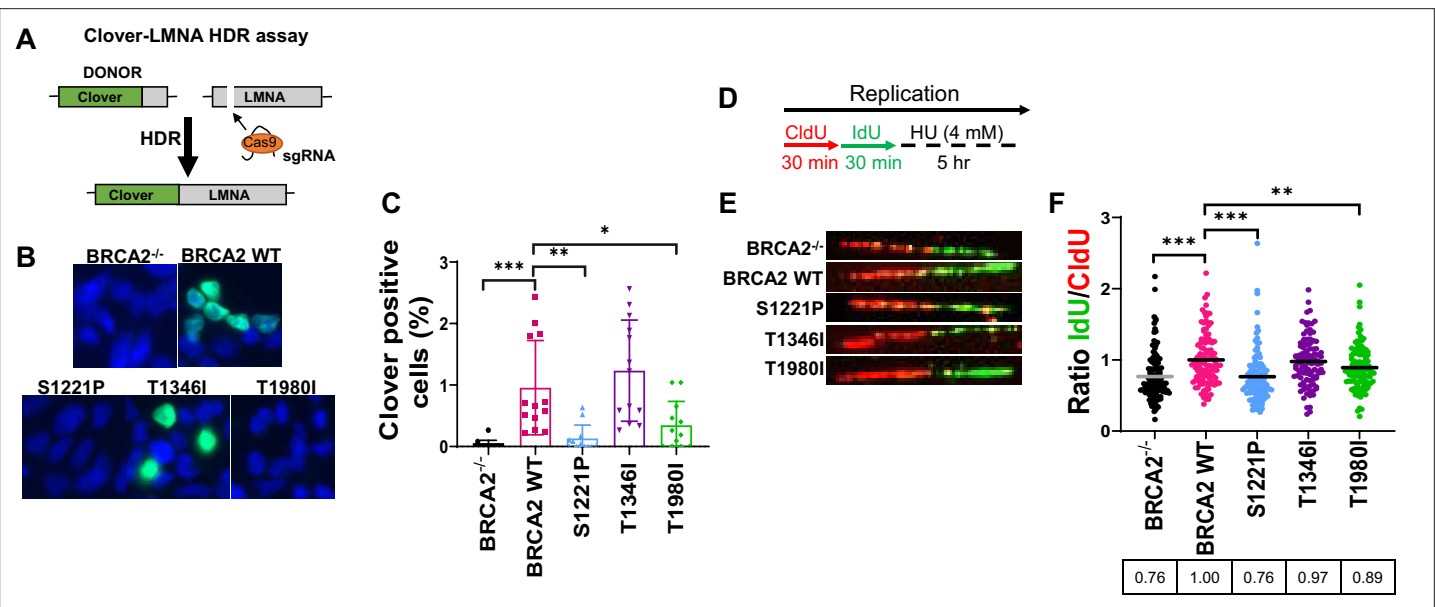

**Figure 3.** S1221P and T1980I exhibit defects in HDR and fork protection. (**A**) Schematic of the Clover-LMNA HDR assay. (**B**) Immunofluorescence images and quantification of Clover-LMNA HDR (green) and DAPI staining to visualize nuclei (blue) 96 hours post-transfection (threshold 0.25). (**C**) Quantification of the percentage of cells Clover positive in 4 independent experiments. (**D**) Schematic of CldU/IdU pulse-labeling followed by a 5 hr hydroxyurea (HU; 4 mM) treatment with representative images (**E**) of CldU (Red) and IdU (Green) replication tracts after HU treatment. (**F**) Dot plot of IdU to CldU tract length ratios for individual replication forks in HU-treated cells. All samples are normalized to BRCA2 WT (ratio = 1, i.e., functional fork protection). The median value of 100 or more IdU and CldU tracts per experimental condition is indicated. p-value <0.05, p-value <0.01, p-value <0.001.

The online version of this article includes the following source data for figure 3:

**Source data 1.** Contains quantification of clover positive cells (*Figure 3C*), images of DNA fibers (*Figure 3E*), DNA fiber analysis (*Figure 3F*).

suppress micronuclei formation, T1980I was unable to do so, and S1221P appeared to exacerbate micronuclei formation (see *Figure 2C* **DAPI panels (right) & quantitation in lower graph**).

## S1221P and T1980I, but not T1346I, are defective in HDR and fork protection

We used a CRISPR-Cas9 based gene-targeting assay to investigate whether S1221P, T1980I, and T1346I affect HDR (*Figure 3A*; *Orthwein et al., 2015*; *Pinder et al., 2015*; *Zhao et al., 2017*). As expected, WT BRCA2 complementation increased the percentage of correctly targeted Clover positive cells at the LMNA locus compared to DLD1 BRCA2$^{-/-}$ cells (*Figure 3B and C*). In contrast, expression of S1221P or T1980I was unable to rescue HDR-mediated gene targeting to the same extent as WT BRCA2 detected as a lower percentage of Clover positive cells (*Figure 3C*). In addition to the canonical HDR functions of loading and stimulating RAD51 nucleation onto resected DNA, BRCA2 has been shown to protect stalled replication forks from nucleolytic degradation by nucleases such as MRE11 (*Schlacher et al., 2011*). Fork protection functionality was investigated using a sequential dual label approach. Cells were labeled with thymidine analogues, CldU and IdU, for 30 min followed by a five hour hydroxyurea (HU) treatment (*Figure 3D*; *Maya-Mendoza et al., 2013*; *Schlacher et al., 2011*). Representative images of individual fibers as well as a dot plot analysis of the IdU/CldU ratio are shown in *Figure 3E and F*, respectively. Our results (*Figure 3F*) confirm previous studies (*Schlacher et al., 2011*) in that DLD1 BRCA2$^{-/-}$ cells have a defect in HU-induced fork protection compared to BRCA2 WT complemented cells (24% reduction). T1346I protected forks similar to WT BRCA2, however, both S1221P and T1980I exhibited deficits (11–24% reduction).

## Individual S1221P and T1980I BRC mutations eliminate binding to RAD51

To interrogate protein interactions, we took advantage of our 2XMBP tagged full-length BRCA2 constructs stably expressed in DLD1 BRCA2$^{-/-}$ cells and performed amylose pull-downs to capture BRCA2 and probe for endogenous RAD51 binding (*Figure 4—figure supplement 1A*). Equal levels of RAD51 were pulled down by WT BRCA2 and each variant despite the presence of individual BRC mutations. We performed the same experiment in a transient transfection approach in 293T cells and observed comparable levels of RAD51 binding across all proteins (*Figure 4—figure supplement 1B*).

We reasoned that full-length BRCA2 S1221P and T1980I variants appear to bind RAD51 in an equivalent manner to the WT protein either due to the insensitivity of our assay conditions or to the presence of the remaining seven wild type BRC repeats masking the defect in a single BRC repeat. We queried a panel of 2XMBP-BRC repeats consisting of individual as well as multiple BRC repeat fragments expressed in 293T cells: BRC1, 2, 4, 7, BRC1-2, BRC3-4, BRC5-6, BRC7-8, BRC1-4, BRC5-8, and BRC1-8, analyzed as above to assess RAD51 binding characteristics (*Figure 4—figure supplement 2A, B*). To our surprise, BRC7-8 bound RAD51 while BRC5-6 and the BRC5-8 region did not. In a previous study, we found that a BRC5-8-DBD fusion protein also failed to bind RAD51 (*Chatterjee et al., 2016*). Conversely, BRC4, BRC1-4, and BRC1-8 fusions with the DBD displayed robust RAD51 binding (*Chatterjee et al., 2016*). Prior yeast two-hybrid analyses established that all BRC repeats interact with RAD51 except for BRC5 and BRC6 in agreement with a separate study demonstrating that GFP-RAD51 binds GST-BRC7-8 (*Wong et al., 1997*; *Yu et al., 2003*). Indeed, we confirmed that BRC7 alone can bind RAD51 under our pull-down conditions (*Figure 4B*, lane 11 and *Figure 4—figure supplement 2B*, lane 15), albeit to a much lesser extent than BRC2 (*Figure 4B*, lane 9).

To circumvent the potential for compensatory BRC repeat interactions with RAD51, we generated a single BRC2 construct incorporating the S1221P variant, and likewise, created a BRC7 construct with the T1980I variant. We included BRC4 T1526A in our analysis, as this variant has been described to interfere with RAD51 binding (*Tal et al., 2009*). The 2XMBP tag alone was used as a negative control in all experiments. By focusing on the individual BRC repeats, we found that both S1221P and T1980I demonstrated no detectable RAD51 interaction in our pull-down analyses in 293T cells (*Figure 4B*, compare lanes 9 & 10 and lanes 11 & 12 and *Figure 4—figure supplement 2B*, lanes 16 & 18). As expected, T1526A disrupts the RAD51 interaction although we observed some residual binding (*Figure 4—figure supplement 2B*, lane 17). To further confirm our results in an orthogonal approach, we synthesized the individual peptides (33 amino acids), conjugated them to aminolink resin, and incubated the BRC peptides with purified RAD51. The results clearly show that wild type BRC2 and

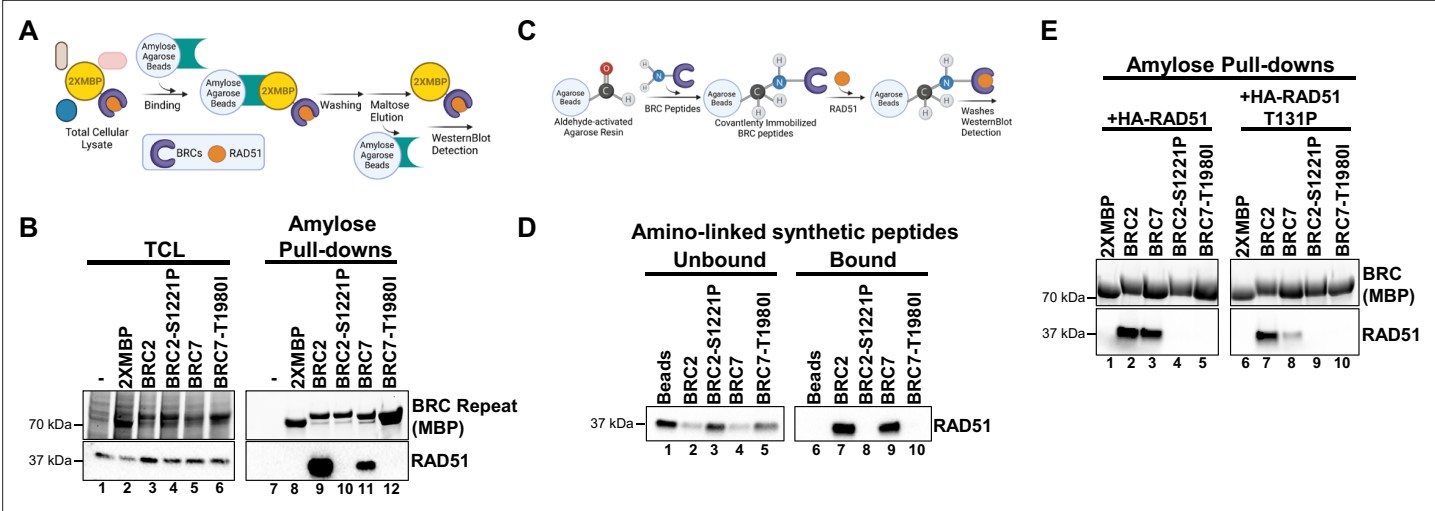

**Figure 4.** BRC2-S1221P and BRC7-T1980I abolish RAD51 binding. (**A**) Schematic of amylose pull-down reaction to detect RAD51 binding. (**B**) Total cellular lysates (TCL) and amylose pull-downs from HEK 293T cells transiently transfected with 2XMBP, 2XMBP-BRC2, 2XMBP-BRC2 S1221P, 2XMBP-BRC7, and 2XMBP-BRC7 T1980I. Western blot: anti-MBP was used to detect 2XMBP-BRC and anti-RAD51 to detect endogenous RAD51. (**C**) Schematic of aminolink-conjugated resin reaction to BRC peptides and pull-down reaction to detect purified RAD51 binding. (**D**) Synthesized peptides BRC2, BRC2 S1221P, BRC7, BRC7 T1980I (35 amino acids) were conjugated to aminolink resin and incubated with purified RAD51. Western blot using RAD51 antibody to detect unbound and bound RAD51 after washes and elution of proteins in laemmli sample buffer. (**E**) Amylose pull-downs in HEK 293T cells transiently transfected with 2XMBP, 2XMBP-BRC2, 2XMBP-BRC2-S1221P, 2XMBP-BRC7, 2XMBP-BRC7 T1980I, wild type HA-RAD51, and T131P HA-RAD51. Anti-MBP was used to detect 2XMBP-BRC proteins. HA antibody was used to detect recombinant RAD51.

The online version of this article includes the following source data and figure supplement(s) for figure 4:

**Source data 1.** Contains original (uncropped) gel images for *Figure 4B, D, E*.

**Figure supplement 1.** S1221P, T1346I, and T1980I full-length BRCA2 proteins bind RAD51.

**Figure supplement 1—source data 1.** Contains band quantification and original (uncropped) gel images.

**Figure supplement 2.** BRC1-2, BRC7-8, BRC2, and BRC7 bind RAD51 whereas BRC2-S1221P and BRC7-T1980I ablate RAD51 binding.

**Figure supplement 2—source data 1.** Contains original (uncropped) gel images.

BRC7 peptides interact with RAD51 (*Figure 4D*, lanes 7 & 9) while the S1221P and T1980I peptides abolish binding (*Figure 4D*, lanes 8 & 10). Pre-incubation of wild type BRC2 peptide, but not the S1221P variant, with purified RAD51 protein prevents binding to BRC4 confirming the stable association of BRC2 with RAD51 and lack of binding by the S1221P variant (*Figure 4—figure supplement 2E*, compare lane 8 and 9).

The BRC repeats are composed of two modules: BRC1-4 mediates binding to free monomeric RAD51 while BRC5-8 binds and stabilizes RAD51-ssDNA complexes (*Carreira and Kowalczykowski, 2011*). To reconcile our pull-down data showing that BRC7 interacts with RAD51 in the absence of pre-assembled filaments on ssDNA, we considered that RAD51, both in purified form and likely in cell extracts, may self-oligomerize resembling a filament that mediates binding to both BRC7 and the BRC7-8 peptides (*Subramanyam et al., 2018*). We addressed this discrepancy by interrogating whether the RAD51 T131P mutant was capable of BRC7 interaction. T131P is located in the Walker A domain of RAD51, alters ATP binding and hydrolysis behavior preventing filament formation, and acts as a dominant negative in the presence of WT RAD51 (*San Filippo et al., 2006*; *Wang et al., 2015*; *Zadorozhny et al., 2017*). We confirmed that T131P does not self-associate (*Figure 4—figure supplement 2D*, lane 8). Indeed, co-incubation of RAD51 T131P with BRC2 exhibited similar binding as wild type RAD51 (*Figure 4E*, compare lanes 2 & 7), whereas binding of T131P to BRC7 was greatly reduced (*Figure 4E*, lane 8). These results reaffirm that BRC2 binds the monomeric form of RAD51 while BRC7 presumably binds oligomers or self-polymerized forms of RAD51. As expected, the S1221P and T1980I variants failed to bind both wild type RAD51 and the T131P mutant (*Figure 4E*, lanes 4–5 and lanes 9–10).

## S1221P and T1980I fail to stimulate RAD51-ssDNA complex formation

To determine whether the lack of RAD51 binding by BRC2-S1221P and BRC7-T1980I alters stimulation/stabilization of a RAD51-ssDNA complex, we utilized an electrophoretic mobility-shift assay (EMSA) under conditions previously described (*Carreira and Kowalczykowski, 2011*). First, we purified BRC2, BRC7, BRC2-S1221P and BRC7-T1980I from HEK293T cells (*Figure 5—figure supplement 1A*). We titrated RAD51 protein to determine that 10 nM of RAD51 is the optimal concentration at which RAD51-ssDNA complex formation is approximately 20–30% (*Figure 5—figure supplement 1B*). Then, RAD51 was incubated with purified BRC2, BRC2-S1221P, BRC7, and BRC7-T1980I proteins followed by the addition of ssDNA (dT40) (*Figure 5A*). As anticipated, increasing concentrations of BRC2 and BRC7 stimulated binding of RAD51 to ssDNA (*Figure 5B*, lanes 3–6, left panel and lanes 3–6, right panel). In contrast, BRC2-S1221P (*Figure 5B*, lanes 7–10, left panel) and BRC7-T1980I (*Figure 5B*, lanes 7–10, right panel) did not stimulate RAD51-ssDNA complex formation even at the highest concentration of BRC proteins (2 µM) tested (see quantitation in *Figure 5C*).

To assess the impact of the BRC repeat proteins on RAD51 filament formation and stabilization, we utilized a longer DNA substrate (167mer) composed of mixed bases. The reactions were conducted in the absence of calcium to maintain RAD51 ATPase dependent turnover (*Bugreev and Mazin, 2004*). BRC2, BRC2-S1221P, BRC7, BRC7-T1980I, and RAD51 were pre-incubated together followed by the addition of the biotinylated ssDNA substrate (*Figure 5D*). The biotin-DNA-RAD51-BRC complexes were then washed extensively, captured on streptavidin magnetic beads, and eluates run on an SDS-PAGE gel. In agreement with the results from our EMSA analyses, BRC2 and BRC7 stabilized RAD51 complex formation on the ssDNA substrate (*Figure 5E*, lanes 5–6 and lanes 11–12) compared to RAD51 alone (*Figure 5E*, lanes 2–3). BRC2-S1221P (*Figure 5E*, lanes 8–9) and BRC7-T1980I (*Figure 5E*, lanes 14–15) showed little to no increase in bound RAD51. Consistent with BRC7 playing a larger role in RAD51-ssDNA complex formation (*Carreira and Kowalczykowski, 2011*), we noted increased levels of RAD51 were stably associated with biotinylated ssDNA in the presence of BRC7 compared to BRC2 (compare lane 12 to lane 6 in *Figure 5E*). Interestingly, residual RAD51-ssDNA binding was observed with BRC7-T1980I (compare lane 15 to lane 3 in *Figure 5E*) perhaps indicative of a binding mode specific to BRC7 that is not completely diminished by the T1980I mutation.

To determine how the BRC variants contribute to RAD51 nucleation at single molecule resolution in real-time, we performed a <u>s</u>ingle <u>m</u>olecule <u>F</u>örster <u>R</u>esonance <u>E</u>nergy <u>T</u>ransfer (smFRET) assay. Our smFRET design placed the donor fluorophore (green) in the ssDNA while the acceptor fluorophore (red) was positioned 16 nucleotides from the donor, at the ss/dsDNA junction (*Figure 5F*). In this configuration, the flexible ssDNA region brings the donor and acceptor molecules into close proximity leading to efficient energy transfer between the FRET pair and corresponds to a high FRET state when plotted as a FRET histogram (*Figure 5F*). Upon RAD51 binding, the ssDNA is extended, resulting in an increase in the distance between the donor and acceptor thereby reducing the efficiency of FRET, and a shift to lower FRET states see example of 400 nM RAD51 (*Figure 5—figure supplement 2*). To observe stimulation of RAD51-ssDNA complexes by the BRC peptides, we utilized a sub-stoichiometric level of RAD51 (20 nM) at which no change in FRET is observed with RAD51 alone (*Figure 5F*). Pre-incubation of 20 nM WT BRC2 with 20 nM RAD51 (1:1 ratio) maintained the same FRET state as RAD51 alone. However, WT BRC7 pre-incubated with RAD51 reduced the high FRET signal ($E_{FRET} = 0.70$) to a lower state ($E_{FRET} = 0.60$) indicating that BRC7 can stimulate RAD51 nucleation or filament formation on the ssDNA substrate. Surprisingly, despite a range of concentrations and molar ratios tested, we were unable to detect any change in FRET mediated by WT BRC2. Our results with BRC2 S1221P and BRC7 T1980I agreed with our gel-based analyses as no change in FRET was observed (*Figure 5F*, *Figure 5—figure supplement 3*). Control experiments with all BRC peptides in the absence of RAD51 confirmed no change in the high FRET state (*Figure 5—figure supplement 2*). Taken together, the data support the hypothesis that S1221P and T1980I harbor a biochemical defect compromising the ability of BRCA2 to bind and stabilize the RAD51 nucleoprotein complex.

## The full-length BRCA2 T1980I protein binds DNA but exhibits impaired stimulation of RAD51-dependent DNA strand exchange activity

To further probe the impact of the S1221P and T1980I mutations on BRCA2 functions, we attempted to purify the full-length proteins (*Figure 6—figure supplement 1A*). Despite extensive troubleshooting, we were unable to purify the full-length S1221P protein due to instability and degradation

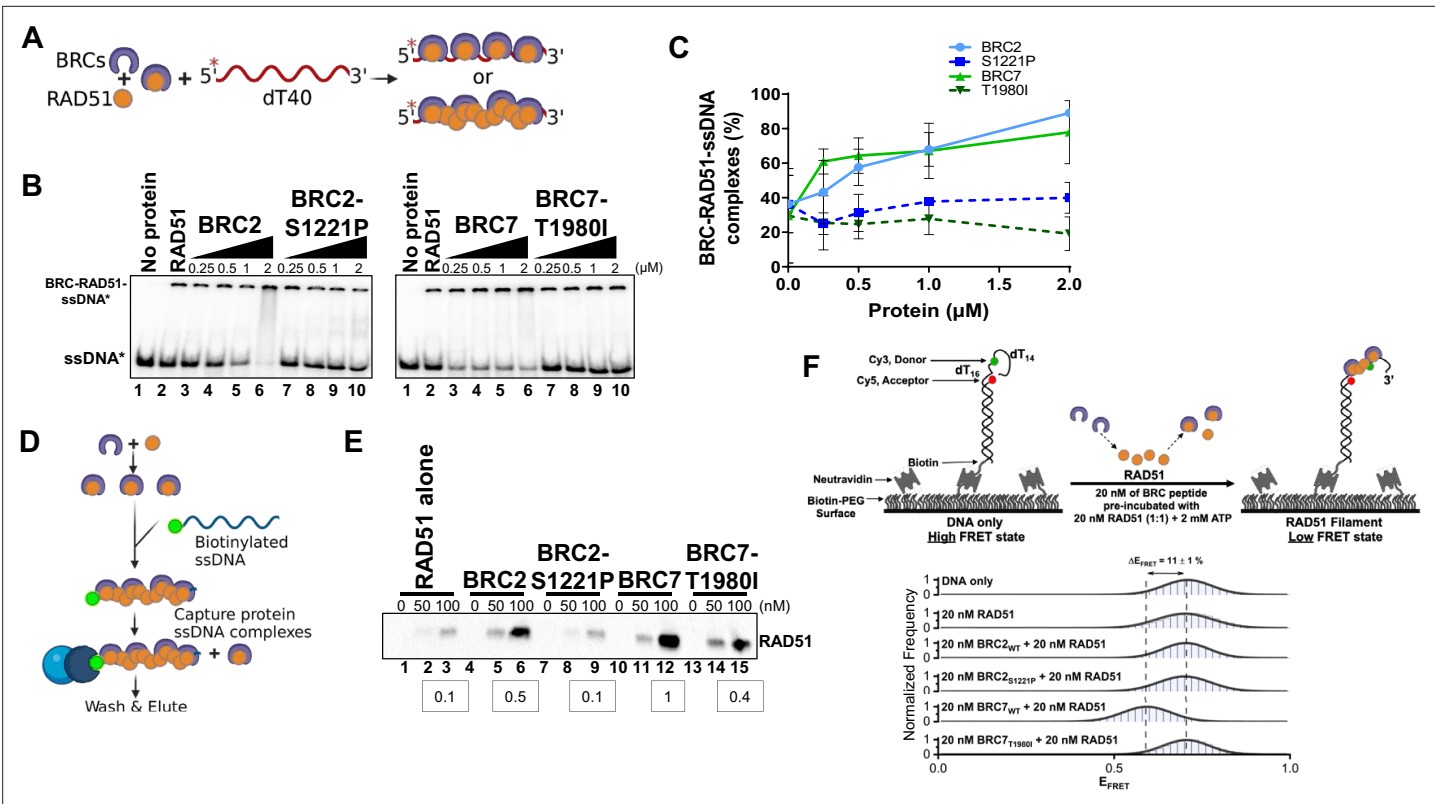

**Figure 5.** BRC2-S1221P and BRC7-T1980I fail to stimulate RAD51-ssDNA complex formation. (**A**) Schematic of reaction to assay BRC repeat stimulation of RAD51-ssDNA complex formation by EMSA. RAD51 was pre-incubated with increasing amounts of BRC protein for 15 min then radiolabeled ssDNA (dT40) was added for 40 min. All reactions were incubated at 37 degrees and visualized on a 6% TAE polyacrylamide gel. (**B**) Autoradiograms of EMSA gels depicting increasing concentration of BRC proteins: BRC2, BRC2-S1221P, BRC7, BRC7-T1980I incubated with 10 nM RAD51 and 400 pM ssDNA (dT40*). Lane 1 is no protein control. Lane 2 is RAD51 alone. (**C**) Quantification of BRC-RAD51-ssDNA complexes calculated from gels shown in B. Error bars represent the S.D. for two biological independent experiments. (**D**) Schematic of biotinylated DNA pull-down assay. Purified BRC proteins were pre-incubated with increasing concentrations of purified RAD51 for 10 min. Biotinylated ssDNA (167-mer) was then added for 10 min to allow nucleoprotein filament formation and captured on magnetic streptavidin beads. The beads were then washed, eluted in sample buffer, analyzed by SDS-PAGE, and bound RAD51 was detected by western blotting using an anti-RAD51 antibody. (**E**) Western blot depicting RAD51 pulled down and eluted from biotin-DNA-BRC-RAD51 complexes. 0, 50, or 100 nM RAD51 was pre-incubated with 80 nM of BRC peptide, incubated with biotin-ssDNA, washed extensively, and eluted. Densitometric quantitation of RAD51 binding at 100 nM is indicated in the boxes below. (**F**) Schematic depicting single-molecule FRET (smFRET) assay with 30-nucleotide 3'-tail ssDNA. Addition of BRC7$_{WT}$ results in an increase in RAD51 binding leading to a transition from high FRET (DNA-only) to medium FRET (BRC7$_{WT}$ and RAD51-bound). Histograms display a shift (ΔE$_{FRET}$ = 11%) upon addition of 20 nM BRC7$_{WT}$ and 20 nM RAD51 that is not observed in BRC7$_{T1980I}$ or in the BRC2 peptides (WT and S1221P). Representative histograms do not include zero FRET values or photobleached portions of the FRET trajectories. A minimum of 250 smFRET trajectories were utilized to generate each histogram.

The online version of this article includes the following source data and figure supplement(s) for figure 5:

**Source data 1.** Contains uncropped gel images (*Figure 5B*), EMSA band quantification (*Figure 5C*), biotin-DNA pulldown band quantification (*Figure 5E*), histogram data (*Figure 5F*).

**Figure supplement 1.** Purified proteins used for biochemical studies.

**Figure supplement 1—source data 1.** Contains original (uncropped) gel images.

**Figure supplement 2.** smFRET control assays highlighting BRC wildtype and mutant proteins (S1221P and T1980I) show no change in FRET efficiency in the absence of RAD51.

**Figure supplement 2—source data 1.** Contains data and traces for single molecule FRET.

**Figure supplement 3.** Representative trajectories corresponding to histograms in *Figure 5F*.

**Figure supplement 3—source data 1.** Contains data and traces for single molecule FRET.

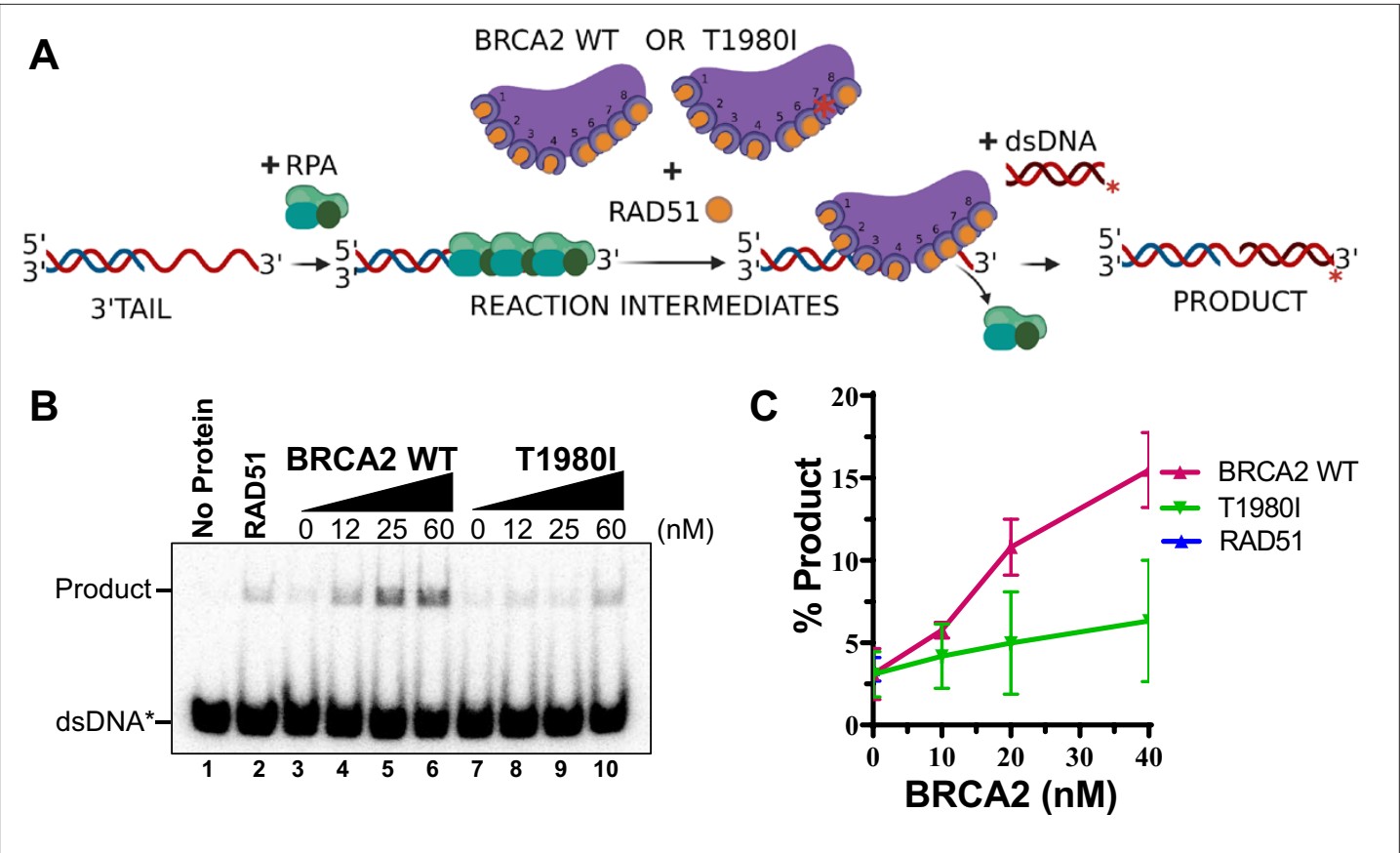

**Figure 6.** RAD51-mediated DNA strand exchange activity is stimulated by increasing amounts of WT BRCA2 but not T1980I. (**A**) Diagram of DNA strand exchange assay. RPA was pre-bound to the 3′ tail DNA substrate for 5 min. Increasing amounts of WT BRCA2 or T1980I protein was then added in combination with RAD51 for 5 min, followed by the addition of radiolabeled donor dsDNA for 30 min. All reactions were incubated at 37 degrees. The reaction was then deproteinized and run on a 6% TAE polyacrylamide gel. (**B**) Autoradiograms of PAGE gels used to analyze the products of DNA strand exchange. Lane 1 is no protein control. Lane 2 is RAD51 in the absence of RPA or BRCA2. (**C**) Quantification of product formation from autoradiogram in (**B**); mean values were plotted. Errors bars represent the S.D. of three biological independent experiments.

The online version of this article includes the following source data and figure supplement(s) for figure 6:

**Source data 1.** Contains original (uncropped) gel image (*Figure 6B*) and band quantitation (*Figure 6C*).

**Figure supplement 1.** Purified full-length BRCA2 WT and T1980I proteins bind a panel of DNA substrates with equivalent efficiencies.

**Figure supplement 1—source data 1.** Contains original (uncropped) gel images and band quantitation.

issues. Purification of full-length T1980I protein was successful (*Figure 6—figure supplement 1A*, lane 3). To evaluate the DNA binding profile of purified T1980I, we performed EMSAs on 3′ Tail, 5′ Tail, ssDNA, and dsDNA substrates (*Figure 6—figure supplement 1B*). T1980I bound all DNA substrates with comparable affinities to the WT BRCA2 protein.

We next evaluated whether stimulation of RAD51-mediated DNA strand exchange was impacted by the T1980I variant. We implemented a strategy utilized previously to define the mediator role of wild type BRCA2: (1) we first incubated a 3′ tail DNA substrate with RPA, (2) followed by the addition of wild type BRCA2, or T1980I, simultaneously with RAD51, and (3) finally, a radiolabeled donor dsDNA was added to initiate the DNA strand exchange reaction (*Jensen et al., 2010* and *Figure 6A*). Strikingly, T1980I was severely impaired for stimulation of RAD51-mediated DNA strand exchange (*Figure 6B and C*). The results corroborate our cell-based findings and define the T1980I biochemical defect as disruption of RAD51 binding leading to a failure to fully stimulate RAD51-ssDNA complex formation and diminished ability to mediate RAD51-dependent DNA strand exchange. Although full-length BRCA2 S1221P protein proved to be intractable to purification, we speculate that stimulation of RAD51-mediated DNA strand exchange activity would be reduced to similar levels due to defects in both RAD51 binding and stabilization of RAD51-ssDNA complexes.

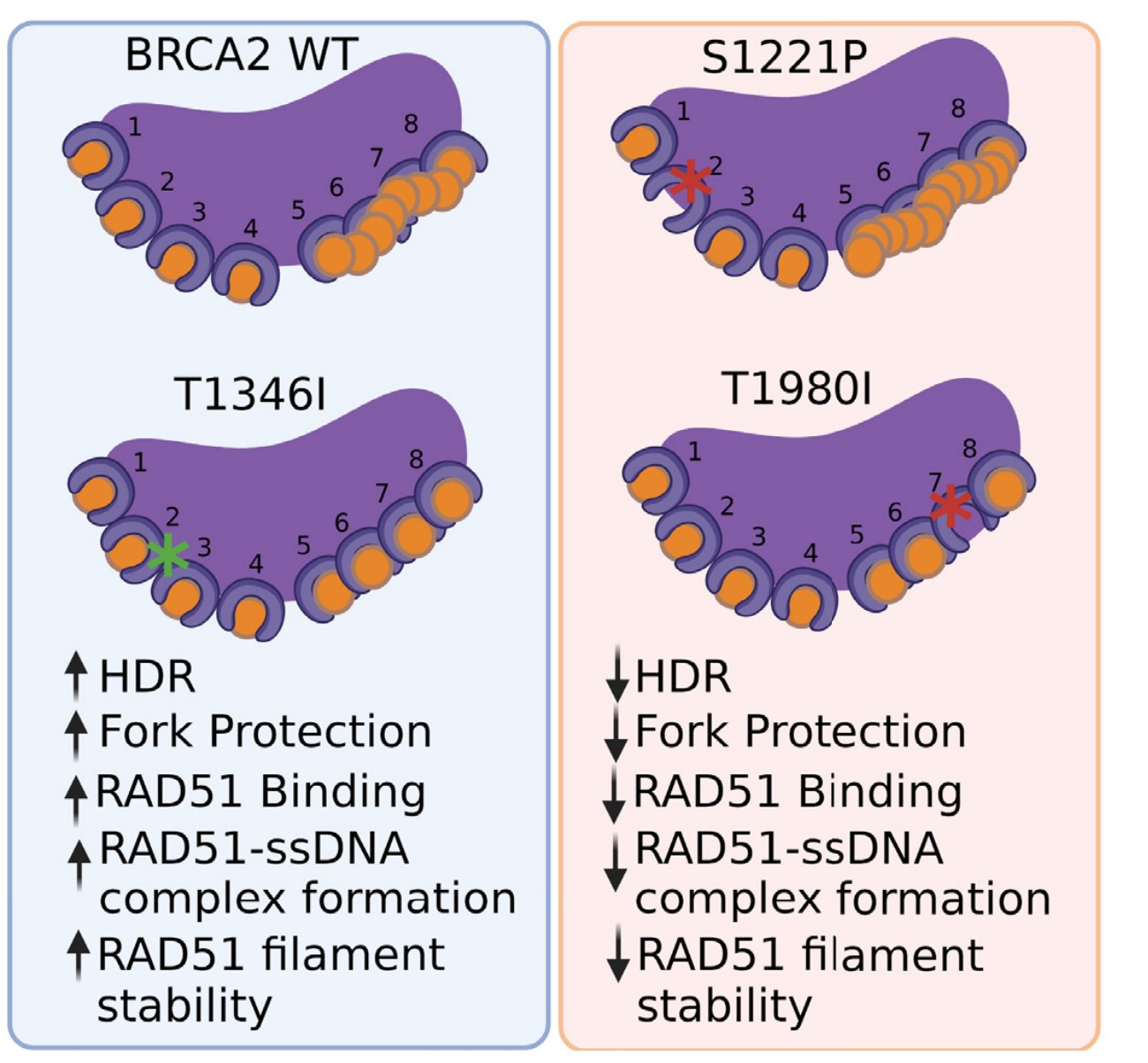

**Figure 7.** BRCA2 missense variants S1221P (BRC2) and T1980I (BRC7), but not T1346I (spacer BRC2-3).
 (1) exhibit decrease in HDR and fork degradation, (2) disrupt RAD51 binding and (3) fail to stimulate RAD51-ssDNA complex formation and RAD51 filament stability.

## Discussion

Through a comprehensive analysis of three independent variants located in the BRC repeats of BRCA2, our goals were to not only identify potentially pathogenic variants with important clinical implications for cancer risk in patients, but to also leverage deleterious variants to uncover the specific functions carried out by individual BRC repeats. Our findings reveal that specific BRC repeat mutations within the context of the full-length BRCA2 protein impair critical functions despite the presence of the remaining functional BRC repeats. Our work provides evidence that the BRC7 repeat stimulates RAD51 nucleation/filament stability in a different manner than the BRC2 repeat. We demonstrate that BRC7 alone can bind RAD51 but only under conditions where RAD51 can self-associate, as in the case of a nucleoprotein filament. Collectively, our results with patient-derived missense variants highlight that single amino acid changes in an individual BRC repeat, within the context of the full-length 3418

amino acid BRCA2 protein, substantively impact functions of BRCA2 related to RAD51 binding and regulation during HDR and fork protection (*Figure 7*).

The eight proximal BRC repeats of human BRCA2 bind RAD51 with varying affinities and facilitate the loading and nucleation of RAD51 onto ssDNA. Each BRC repeat comprises a highly conserved motif, FxTASGK, crucial for BRCA2–RAD51 interactions (*Figure 1A and B*). Our study has shown that site-specific variants in individual BRC repeats alter the biochemical functions of BRCA2 leading to cellular defects in response to DNA damage providing functional evidence for pathogenicity. Specifically, we found that missense variants located at the third amino acid position in the Fx**T**ASGK motif, S1221P (BRC2) and T1980I (BRC7): (1) disrupt RAD51 binding; (2) do not fully rescue survival in response to chemotherapeutics; (3) diminish RAD51 foci formation; (4) exhibit fork protection defects; (5) fail to stimulate RAD51-ssDNA complex formation; and (6) in the case of T1980I, binds DNA but does not stimulate RAD51-dependent DNA strand exchange (*Figure 7*). Our findings emphasize the importance of the serine and threonine residues in the conserved Fx**T**ASGK motif located within each BRC repeat (*Lo et al., 2003*). Similar defects were reported for the BRC4 mutant T1526A and a BRC3 mutant (*Carreira et al., 2009*; *Davies et al., 2001*). The Fx**T**ASGK motif has been noted in other proteins such as RECQL5 and MMS22L, and mutations that disrupt RAD51 binding downregulate HDR and replication fork protection/stability (*Islam et al., 2012*; *Piwko et al., 2016*).

To the best of our knowledge, we are the first to characterize the BRCA2 T1346I variant, within the context of the full-length protein, found somatically mutated in a colorectal tumor specimen. BRCA2 missense variants located between BRC repeats have been classified as benign in ClinVar and the Breast Information Core (BIC) databases through bioinformatic prediction and familial linkage analysis. We were unable to uncover any defects in the T1346I variant located between BRC2 and BRC3; however, more studies will be required to determine whether other pathogenic variants exist in the spacer regions between the BRC repeats.

Despite tremendous progress made in the past two decades delineating the explicit functions of the BRCA2 protein, many questions remain regarding the biological significance of multiple RAD51 binding modules located in the BRC repeats and carboxy terminus (S3291) of BRCA2. Furthermore, why does human BRCA2 possess eight BRC repeats whereas organisms such as *Ustilago maydis* and *Caenorhabditis elegans* contain only one BRC repeat (*Kojic et al., 2002*; *Martin et al., 2013*)? The human BRC repeats have been found to interact with RAD51 with varying affinities based on yeast two-hybrid, co-immunoprecipitation, and pull-down assays using tagged purified proteins (*Carreira and Kowalczykowski, 2011*; *Chatterjee et al., 2016*; *Chen et al., 1999*; *Chen et al., 1998*; *Esashi et al., 2005*; *Thorslund et al., 2007*; *Wong et al., 1997*). The emerging consensus is that BRC1, 2, 3, and 4 bind monomeric RAD51 with high affinity, whereas BRC5, 6, 7, 8 bind weakly. However, BRC5, 6, 7, and 8 significantly stimulate RAD51-ssDNA complex formation implying specific interactions and stabilization of the RAD51 nucleoprotein filament (*Carreira and Kowalczykowski, 2011*; *Chatterjee et al., 2016*). Lastly, the detailed structure, orientation, and regulation of the eight human BRC repeats within the framework of the full-length BRCA2 protein remain unknown.

We confirmed that BRC1-2, BRC3-4, BRC7-8 and individual BRC1, 2, 4, and 7 repeats bind RAD51 (*Figure 4—figure supplement 2A, B*). However, as noted in our previous study (*Chatterjee et al., 2016*), BRC5-8 and BRC5-6 do not bind RAD51 in a non-filament form (Figure S5A). Our results agree with prior conclusions from yeast two-hybrid experiments and pull-down interaction studies with GFP-RAD51 in human cells (*Wong et al., 1997*; *Yu et al., 2003*). BRC5 and BRC6 are the least conserved repeats potentially explaining the lack of RAD51 binding (*Bignell et al., 1997*; *Lo et al., 2003*). The lack of BRC7 binding to RAD51 T131P (*Figure 4E*) lends support to the hypothesis that the BRC5-8 module interacts specifically with the RAD51 filament, in agreement with our previous data and prior publications (*Carreira and Kowalczykowski, 2011*; *Chatterjee et al., 2016*). Intriguingly, BRC repeats 1–4 inhibit the ssDNA-dependent ATPase activity of RAD51 whereas BRC repeats 5–8 appear to have no role in this function (*Carreira and Kowalczykowski, 2011*). Conversely, BRC repeats 5–8 stimulate RAD51-ssDNA complex formation through an unknown mechanism (*Carreira and Kowalczykowski, 2011*). Our prior work demonstrated that BRC5-8 tethered to the DBD of BRCA2 provided greater functionality than BRC1-4-DBD, and even surprisingly BRC1-8-DBD, in both cell-based and biochemical HDR assays. It remains to be determined why BRC5-8-DBD, lacking the N-terminus and BRC1-4 repeats, demonstrated higher activity than other BRC-DBD fusion proteins. Interestingly, our smFRET data (*Figure 5F*) establishes a clear distinction between the two categories of BRC repeats,

exemplified by BRC2 and BRC7. Using sub-stoichiometric concentrations of RAD51, we demonstrated that BRC7, unlike BRC2, stabilized RAD51-ssDNA complexes. Little is known regarding the regulatory and structural roles of each BRC repeat, and thus, our smFRET data provides further evidence for differential effects mediated by individual BRC repeats on RAD51 nucleation and stability at single molecule resolution. As our smFRET studies directly measure changes in the distance between two fluorophores at fixed positions along the DNA substrate, we cannot rule out the possibility that BRC7 may facilitate a change in the RAD51-ssDNA conformation unrelated to nucleation/filament formation leading to the lower FRET state. However, given the established role of BRCA2 in facilitating RAD51-ssDNA stability, we favor the conclusion that BRC7 is indeed stimulating RAD51 nucleation and/or filament formation. Future studies using site-directed mutagenesis to systematically inactivate RAD51 binding in each of the BRC repeats, similar to the work described here for patient-derived missense variants, should clarify how the two BRC modules contribute to HDR functions.

VUS described in the literature are often miscategorized as 'tumor-associated' with unknown impact on protein function, and subsequently, unknown association with tumorigenicity. Reliance on software prediction algorithms such as SIFT or PolyPhen to predict pathogenicity, rather than functional assays, may or may not result in correct classification. Uncertainties surrounding VUS interpretation have become a substantial clinical challenge for physicians and genetic counselors attempting to make critical treatment decisions or counsel patients about future cancer risk. Adoption of functional laboratory-based assays into the clinic has been slow due to low throughput, the expertise required for interpretation, and the need for extensive validation utilizing known pathogenic variants. Nonetheless, considerable progress has been made studying variants located in the DBD domain of BRCA2 using the DR-GFP reporter assay or rescue of mouse embryonic stem cell lethality (*Biswas et al., 2011*; *Biswas et al., 2012*; *Biswas et al., 2020*; *Carvalho et al., 2007*; *Farrugia et al., 2008*; *Guidugli et al., 2014*; *Guidugli et al., 2013*). Expanding the repertoire of variants to other domains, as done here for the BRC repeats, will be vital to build a comprehensive molecular understanding of how specific defects lead to BRCA2 dysfunction. The utility of patient-derived missense variants with accompanying clinical data lies not only in their potential to improve cancer risk prediction models but to also provide clinically relevant mutations that can further elucidate the underlying biology of the BRCA2 protein. Importantly, rigorous biochemical and genetic assays hold promising potential to differentiate pathogenic from benign BRCA2 variants. Moreover, functional assays may be the only recourse to advise patients with rare variants concerning their future cancer risk.

Given the total number of BRCA2 variants that currently exist (15,079 in ClinVar accessed 2022-07-12) and will continue to increase as genome sequencing efforts become incorporated into clinical care, large-scale studies will be necessary to evaluate functionality. However, the specific functions of BRCA2 required for tumor suppression have not been rigorously identified. Our assumption that canonical HDR functions (RAD51 loading/filament stabilization) underlie BRCA2's role in tumor suppression seems likely but requires definitive proof. BRCA2 is a large, complex protein that paradoxically is required for cellular viability, and yet, its loss in somatic human cells drives tumor initiation and/or progression. For large-scale testing of BRCA2 VUS to be successful in predicting cancer risk, we need to carefully consider which specific functions impart tumor suppression. Will the assay use in large-scale studies measure rescue of viability, HDR, fork protection, gap suppression, or some other attribute? To date, there is no one assay that directly measures the tumorigenic capacity of BRCA2 mutations as has been done for certain oncogenes. We favor the idea that in-depth biochemical and cellular analysis of patient derived BRCA2 variants with known clinical outcomes will help elucidate the features required for tumor suppressor functions. Because most human cell lines are not viable in the absence of the BRCA2 protein, the unique advantage of the DLD1 BRCA2$^{-/-}$ cell system is that it provides us with a cellular tool to perform complementation studies with both benign and deleterious variants. The variants can be assessed for response to clinically relevant chemotherapeutic agents such as cisplatin and PARP inhibitors. Leveraging the transfection efficiency and overexpression capabilities of human 293T cells, we can then purify the full-length variant proteins and assess biochemical functions to understand the altered molecular mechanisms (*Jensen et al., 2010*).

Clinical integration of functional assays into the genetic counseling setting is an important goal but should be undertaken with caution until we fully understand how specific variants impact the tumor suppressor functions of BRCA2. Our study demonstrated that novel pathogenic variants exist not only in the DBD domain of BRCA2 but also in the BRC region leading to defects in RAD51 binding, activity,

and subsequent HDR deficiencies. Mechanistic studies leveraging patient variants will continue to reveal the many functions of BRCA2.

## Materials and methods

### Sequence alignment and molecular modeling of BRC repeats

BRCA2 amino acids sequences from 9 different organisms were obtained from Uniprot Knowledge-base database https://www.uniprot.org/ (*Consortium, 2011*). Alignments were done with ClustalX (*Larkin et al., 2007*) http://www.ch.embnet.org/software/ClustalW.html, and Bioedit html (*Hall, 1999*). SnapGene software was used for BRC2, BRC4 and BRC7 sequence alignment (>60% threshold for shading). The UniprotKB codes of the sequences from different organisms were: **P51587** *Homo sapiens (Human)*; Q**9W157** *Drosophila melanogaster (Fruit fly)*; **P97929** *Mus musculus (Mouse)*; **Q864S8** *Felis catus (Cat)*; **O35923** *Rattus norvegicus (Rat)*; **A5A3F7** *Strongylocentrotus purpuratus (Purple sea urchin)*; **A4ZZ89** *Monodelphis domestica (Gray short-tailed opossum)*; **Q8MKI9** *Canis lupus familiaris (Dog) (Canis familiaris)*; **A4ZZ90** *Xenopus tropicalis (Western clawed frog) (Silurana tropicalis)*.

Models are computed by the SWISS-MODEL server homology modelling pipeline (*Waterhouse et al., 2018*), which relies on ProMod3 (*Studer et al., 2021*), an in-house comparative modelling engine based on OpenStructure (*Biasini et al., 2013*). Swiss-Model Expasy server was used to model BRC2 and BRC7 against BRC4 in the PDB structure 1N0W (*Pellegrini et al., 2002*; *Waterhouse et al., 2018*). The modeled BRC2 and BRC7 structures had QMEAN Z-scores greater than –4 (0.06 and –1.42 respectively) and were considered good quality models.

The PyMOL Molecular Graphics System, Version 1.3 Schrödinger, LLC. was used to visualize the BRC repeat aligned with the 1N0W structure to look at predicted interactions with RAD51. BRC residues were mutated using the mutagenesis tool and rotamers with a frequency score greater than 10% were analyzed and the least number of clashes were visualized.

### Constructs

Point mutations S1221P (3663 bp), T1346I (4038 bp), and T1980I (5940 bp) were cloned into the pBluescript BRCA2 (1–5286 bp) and pUC57 BRCA2 (2141–9117 bp) sequences, respectively, via site-directed mutagenesis. The BRCA2 segments were then subcloned into the phCMV1 mammalian expression vector using NotI and EcoRV restriction enzymes for the S1221P and T1346I insert and SbfI and AgeI for the T1980I insert. We verified the putative recombinant clones through restriction digestion and sequencing analysis. The previously described 2XMBP tag (*Jensen et al., 2010*) was placed in-frame at the N-terminus of all proteins separated by an Asparagine linker and the PreScission Protease cleavage sequence.

### Cell culture

All culture media was supplemented with 10% fetal bovine serum (FBS). HEK293T cells were cultured in DMEM *Jensen et al., 2010*; DLD1 cells were cultured in RPMI (*Hucl et al., 2008*) (Horizon Discovery). Transient transfections were carried out with Turbofect (Thermo Scientific) (2 µg of DNA, 6-well plate) in HEK293T cells and with JetOptimus (Polyplus Transfection) in DLD1 cells following the manufacture's protocol. Calcium phosphate transfection (25 µg of DNA per 15 cm² plate, see BRCA2 purification section) was used for large scale purifications in HEK293T cells (*Jensen et al., 2010*). All cell lines were tested regularly for mycoplasma (Mycoalert, Lonza) and authenticated through ATCC STR profiling.

### Generation of stable cell lines

Human colorectal adenocarcinoma DLD-1 BRCA2$^{-/-}$ cells (Horizon Discovery, originally generated by *Hucl et al., 2008*) were stably transfected with 2 µg of DNA using Lipofectamine3000 (Invitrogen). After 48 hr, the cells were trypsinized and diluted 1:2, 1:4, and 1:8 into 100 mm plates containing 1 mg/mL G418. Single cell colonies were picked into 96-well plates and subsequently cultured into 24-well plates, 12-well plates, and 6-well plates. Positive clones were isolated, and protein expression was detected by western blot and immunofluorescence analyses.

**Table 1.** Amino acid sequences of synthesized peptides.

| PEPTIDES | AMINO ACID SEQUENCES |
| --- | --- |
| BRC2 | NEVGFRGFYSAHGTKLNVSTEALQKAVKLFSDIEN |
| BRC2-S1221P | NEVGFRGFY**P**AHGTKLNVSTEALQKAVKLFSDIEN |
| BRC7 | SANTCGIFSTASGKSVQVSDASLQNARQVFSEIED |
| BRC7-T1980I | SANTCGIFS**I**ASGKSVQVSDASLQNARQVFSEIED |

## Western blots and amylose pulldowns

Human embryonic kidney HEK293T cells 70% confluent in 6 well plates were transiently transfected with 0.5 µg or 1 µg of the phCMV1 mammalian expression vector containing a 2XMBP fusion to the full-length or partial cDNA of BRCA2 using TurboFect reagent (Thermo Scientific) (*Jensen et al., 2010*). 0.5 µg of 2XMBP tag alone or untransfected cells were used as negative controls. The cells were lysed 48 hr after transfection in 100 µL of lysis buffer: 50 mM HEPES (pH 7.5), 250 mM NaCl, 5 mM EDTA, 1% Igepal CA-630, 3 mM $MgCl_2$, 10 mM DTT and protease inhibitor cocktail (Roche). Cell extracts were batch bound to amylose resin (NEB) for 2 hr to capture the 2XMBP tagged BRCA2 proteins. Total cellular lysate aliquots were taken before batch binding for protein expression analysis. Total cellular lysates and amylose pulldown samples were run on a 4–15% gradient SDS-PAGE TGX stain-free gel (Bio-Rad 456–8086) and transferred to an Immobilon-P membrane (Merck Millipore IPVH00010) in 1 X Tris/glycine buffer (diluted from 10 X Tris/glycine buffer, Bio-Rad 161–0771). The membrane was blocked in 5% milk in 1 X TBS-T (diluted from 10 X TBS-T: 0.1 M Tris base, 1.5 M NaCl, 0.5% Tween-20). Washes and antibody incubations were done with 1 X TBS-T. Primary mouse antibodies against MBP (NEB E8032L, 1:5000) and RAD51 (Novus Biologicals 14b4, 1:1000) and primary rabbit antibody against BRCA2 (Abcam, ab27976) were used for western blotting. Membranes were incubated with secondary mouse and rabbit antibodies (HRP-conjugated, Santa Cruz Biotechnology sc-516102 and sc-2004, respectively). The western blots were visualized using Clarity Western ECL substrate (Bio-Rad 170–5061) for three minutes and scanning with a ChemiDocMP imaging system (Bio-Rad). For amylose pull-down assays in DLD1 stable cells, three 15 $cm^2$ plates were used.

BRC2, BRC7, BRC2-S1221P and BRC7-T1980I synthetic peptides (see *Table 1* below for amino acid sequences) were ordered from Pierce, resuspended in BRCA2 storage buffer (50 mM HEPES pH 8.2, 450 mM NaCl, 0.5 mM EDTA, 10% Glycerol, 1 mM DTT). In vitro amylose pull-down assays were performed in binding buffer 'B': 50 mM HEPES (pH 7.5), 250 mM NaCl, 0.5 mM EDTA, and 1 mM DTT. Purified 2XMBP fusion proteins (2 µg) were incubated with 1 µg purified RAD51 for 30 min at 37 °C and then batch bound to 30 µL of amylose resin for 1 hr at 4 °C. The complexes were washed with buffer B containing 0.5% Igepal CA-630/0.1% Triton X-100, eluted with 30 µL of 10 mM maltose in buffer B, Laemmli sample buffer was added, samples were heated at 54 °C for 4 min, and loaded onto a 4–15% gradient SDS-polyacrylamide gel (Bio-Rad TGX Stain-Free gel). The gel was run for 1 hr at 130 Volts. The proteins were visualized using Stain-Free imaging on a ChemiDocMP system (Bio-Rad). RAD51 protein was visualized by staining with SyproOrange (Invitrogen). The Stain-Free protein bands were quantified using Image Lab software (version 6.1 Bio-Rad).

For competition assays, synthetic BRC peptides were pre-incubated with RAD51 for 30 min. In parallel, 2XMBP fusion proteins expressed in HEK293T cells were pulled down with amylose resin for 30 min as described above. The 2XMBP fusion proteins bound to amylose resin were washed with buffer B containing 1 M NaCl to disrupt non-specific or weak protein interactions.

## Immunofluorescence imaging

Stable cell clones generated from DLD-1 $BRCA2^{-/-}$ cells were grown on coverslips at $10^5$ cells/well in a 24-well plate for 24 hr. Cells were washed twice with 1 X PBS, fixed in 1% paraformaldehyde-2% sucrose in 1 X PBS for 15 min at room temperature, washed twice with 1 X PBS, permeabilized with methanol for 30 min at –20 °C, then washed two more times with 1 X PBS, and finally incubated with 0.5% triton in PBS for 10 min. Samples were blocked with 5% BSA in 1 X PBS for 30 min at room temperature followed by subsequent incubation with primary antibodies against MBP (NEB E8032L, 1:200) and RAD51 (Proteintech 14961–1-AP, 1:100 or Abcam ab63801) in 5% BSA-0.05% TritonX-100 at 4 °C overnight. The next day, cells were washed three times with 1 X PBS and incubated

with goat anti-rabbit and anti-mouse secondary antibodies conjugated to the fluorophores Alexa-488 and Alexa-546 (Thermo Fisher Scientific A11034 and A11003, respectively; 1:1000). Coverslips were washed three times with 1 X PBS, incubated with 30 nM DAPI for 5 min and mounted on slides with FluorSave reagent (Calbiochem 345789). Immunofluorescence images were taken using a Keyence BZ-X800E Fluorescent Microscope with a 40 X or 60 X objective lens. Cells were either untreated (control) or irradiated at 12 Gy using an X-Rad 320 Biological Irradiator and cells were collected at 6 and 30 hr post irradiation for immunofluorescence imaging.

## Clonogenic survival assay

Stable cell clones generated from DLD-1 BRCA2$^{-/-}$ cells were serially diluted and seeded into six-well plates at concentrations of 100 and 500 cells per well in triplicate for plating efficiency. Simultaneously, cells were seeded for treatment in six-well plates at 1000 and 10,000 cells per well in triplicate. Twenty-four hours after seeding, cells were treated with the indicated concentrations of Mitomycin C (1.5 mM stock in water), Cisplatin (100 mM stock in DMSO) for 1 hr in serum-free media and Olaparib (50 mM stock in DMSO) or BMN (10 nM stock in DMSO) for durations of 24 hr. Following treatment, media was aspirated, and cells were washed with 1 X PBS and re-fed with fresh media containing FBS. Cells were cultured for 14 days to allow colony formation, after which they were stained with crystal violet solution (0.25% crystal violet, 3.5% formaldehyde, 72% methanol). Colonies containing 50 or more cells were scored and surviving fractions were determined.

### Clover Lamin A HDR assay

Cells grown on coverslips in six-well plates (5x10$^5$ cells/well) were transfected with sgRNA plasmid targeting Lamin A (pX330-LMNA-gRNA1, addgene 122507) and donor plasmid (pCR2.1 Clover-LMNA Donor, addgene 122507) with 2.4 μg and 0.6 μg of DNA, respectively, using JetOptimus (Polyplus). Single vectors were transfected as negative controls. Cells were imaged alive 96 hr post-transfection using a Keyence BZ-X800E Fluorescent Microscope or were fixed with 1% PFA/2% Sucrose for 10 min and permeabilized with PBS-0.5% Triton buffer for 5 min. Gene targeting efficiency was determined by counting with Cell Profiler (Positive nuclei pipeline, threshold 0.25 for clover channel intensity) the percentage of Clover-positive cells using DAPI (Thermo Fisher 40 X objective, fixed cells) or NucRed Live 647 ReadyProbes Reagent (Thermo Fisher 20 X objective, live cells). Four independent experiments were performed and at least 500 cells were counted in each experiment. Data was plotted and represented in Graph Pad version 9.

### DNA fiber combing (DOI: 10.1016 /j.xpro.2022.101371)

Stable cell clones generated from DLD-1 BRCA2$^{-/-}$ cells were grown in 6 cm plates overnight and were 30% confluency at the time of treatment. Cells were treated with 100 μM CldU in cell culture media, for 30 min, washed three times with PBS and treated with 100 μM ldU for 30 min, followed by three PBS washes. Cells were then treated with 4 mM hydroxyurea (HU) in water, for 5 hr. After the 5 hr treatment, 1.5x10$^5$ cells were harvested, washed in PBS and resuspended in trypsin. 1.2% low melting agarose was added to each sample and an agarose plug was allowed to solidify using a mold at 4 °C for 1 hr. Plugs were then incubated overnight in ESP buffer (10% (w/v) Sarcosyl in 0.5 M pH = 8 EDTA with 20 mg/mL proteinase K) at 50 °C. The next day plugs received 3 one hour washes followed by one 3.5 hr wash in TE10.1 buffer (10 mM Tris-HCL, 1 mM pH = 8 EDTA). Plugs were immersed in 0.5 M MES pH = 7.4 solution and incubated for 20 min at 68 °C, followed by 10 min at 42 °C. β-agarase was added to each sample and incubated overnight at 42 °C. Each DNA solution was added to a disposable reservoir containing 0.5 M MES pH = 7.4. A pre-coated, silanized glass coverslip was submerged into each fibers solution and removed at a slow constant rate. These fibers were stretched and straightened while binding the coverslip through a combing procedure using the FiberComb Molecular Combing System (Genomic Vision). Combed coverslips were then incubated at 65 °C for 2 hr. Fiber coated coverslips were then submerged in a denaturing solution (0.5 M NaOH, 1 M NaCl) for 8 min, washed three times in PBS and dehydrated in 70% and then 100% ethanol. Coverslips were blocked with 5% BSA in PBS for 30 min at 37 °C. Each coverslip was incubated with primary antibodies diluted in 5% BSA for 1 hr. Primary antibodies used were mouse anti-BrdU (BD Biosciences 347580, 1:10) and rat anti-BrdU (Abcam ab6326, 1:40). Secondary antibodies used were goat anti-rat Cy5 (Abcam ab6565, 1:100) and sheep anti-mouse Cy3 (Sigma C2181, 1:100) and were incubated for

**Table 2.** Amino acid sequences of BRC purified proteins.

| CONSTRUCTS | AMINO ACID SEQUENCE |
| --- | --- |
| 2xMBP BRC2 | YLTDENEVGFRGFYSAHGTKLNVSTEALQKAVKLFSDIENISEETSAEVHPISL* |
| 2xMBP BRC2-S1221P | YLTDENEVGFRGFY**P**AHGTKLNVSTEALQKAVKLFSDIENISEETSAEVHPISL* |
| 2xMBP BRC4 | RDEKIKEPTLLGFHTASGKKVKIAKESLDKVKNLFDEKEQGTSEI* |
| 2xMBP BRC4-T1526A | RDEKIKEPTLLGFH**A**ASGKKVKIAKESLDKVKNLFDEKEQGTSEI* |
| 2xMBP BRC7 | GKLHKSVSSANTCGIFSTASGKSVQVSDASLQNARQVFSEIEDSTKQ* |
| 2xMBP BRC7-T1980I | GKLHKSVSSANTCGIFS**I**ASGKSVQVSDASLQNARQVFSEIEDSTKQ* |

30 min. All antibody incubations were at 37 °C in a wet chamber. Coverslips were protected from the light and mounted using Prolong Diamond Antifade (P36970). Fluorescence images were taken using a Keyence BZ-X800E Fluorescent Microscope with a 60 X oil objective lens. Using ImageJ software, the length of each DNA fiber tract was measured and a ratio of IdU/CldU calculated, with a minimum of 100 fibers (Idu and CldU tract) measured for each sample (for further details see *Moore et al., 2022*).

## Protein purification (doi: 10.1038/nature09399 and DOI: 10.1007/978-1-4939-0992-6_17)

2XMBP- BRC2, BRC7, S1221P-BRC2, T1980I-BRC7 or BRCA2 wild-type, S1221P, T1346I, and T1980I (see *Table 2* below for amino acid sequences) were purified as described for the purification full-length BRCA2 protein (*Jensen et al., 2010*; *Lahiri and Jensen, 2021*). The phCMV1 constructs with human BRCA2 cDNA and 2XMBP tag were transiently transfected using CaPO$_4$ precipitation into HEK293T cells and harvested 30 hr post-transfection. Cell extracts were batch bound to amylose resin overnight. Bound proteins tagged with 2XMBP were eluted with 100 mM maltose and 8 mM glucose, loaded onto a HiTrap Q column, and step eluted at 450 mM NaCl. Fractions with peak protein concentrations were collected and snap frozen in liquid nitrogen and stored at –80 °C. Selected fractions were combined and concentrated with an Amicon Pro Purification system (10 kDa, Millipore). All samples were stored at –80 °C. Purified proteins were visualized on a 4–15% gradient SDS-PAGE TGX Stain-Free gel (Bio-Rad 456–8086) and then stained with Coomassie blue (Bio-Rad 1610786). Protein preparations were quantified by running varying levels of protein on a Stain-Free gel and quantifying band intensity with Bio-Rad Image Lab software. Samples were also quantified at an absorbance of 280 nm using a NanoDrop spectrophotometer (for further details see *Jensen, 2014*; *Jensen et al., 2010*). RPA and RAD51 were purified as described previously (*Anand et al., 2018*; *Subramanyam and Spies, 2018*).

**Table 3.** Oligonucleotide sequences.

| Oligonucleotides | Sequence |
| --- | --- |
| RJ-167-mer | 5′-CTG CTT TAT CAA GAT AAT TTT TCG ACT CAT CAG AAA TAT CCG TTT CCT A TA TTT ATT CCT ATT ATG TTT TAT TCA TTT ACT TAT TCT TTA TGT TCA TTT TTT ATA TCC TTT ACT TTA TTT TCT CTG TTT ATT CAT TTA CTT ATT TTG TAT TA TCC TTA TCT TAT TTA-3′ |
| RJ-5′Tail-167mer | 5′-ATT TAT TCT ATT CCT CTT TAT TTT CTC TGT TTA TTC ATT TAC TTA TTT TGT A TT AAT TTC CTA TAT TTT TTA CTT GTA TTT CTT ATT CAT TTA CTT ATT TTG TAT T AT CCT TAT TTA TAT CCT TTC TGC TTT ATC AAG ATA ATT TTT CGA CTC ATC A GA AAT ATC CG-3′ |
| RJ-167-mer complementary | 5′-TAA ATA AGA TAA GGA TAA TAC AAA ATA AGT AAA TGA ATA AAC AGA GAA AAT AAA GTA AAG GAT ATA AAA AAT GAA CAT AAA GAA TAA GTA AAT GAA T AA AAC ATA ATA GGA ATA AAT ATA GGA AAC GGA TAT TTC TGA TGA GTC GAA AAA TTA TCT TGA TAA AGC AG-3′ |
| RJ-PHIX-42–1 | 5′-CGG ATA TTT CTG ATG AGT CGA AAA ATT ATC TTG ATA AAG CAG-3′ |

## Electrophoretic mobility shift assay

RAD51 (10 nM) was pre-incubated with 2XMBP BRC2, BRC2-S1221P, BRC7 and BRC7-T1980I peptides at the indicated concentrations (0, 0.25, 0.5, 1, 2 μM) for 15 min, followed by addition of ssDNA (Oligo dT40 labeled with $^{32}$P at the 5' end, 400 pM) in buffer 25 mM TrisOAc (pH 7.5), 10 mM MgCl$_2$, 2 mM CaCl$_2$, 0.1 μg/μL BSA, 2 mM ATP, and 1 mM DTT. The mixture was incubated 45 min at 37 °C, as indicated.

The following oligonucleotides (see *Table 3* below) were utilized to test BRCA2 WT and T1980I full-length binding. RJ-167-mer and RJ-5'Tail-167mer were radiolabeled with $^{32}$P at the 5'-end using T4 Polynucleotide Kinase (NEB). To generate the 3' Tail and dsDNA DNA substrates, RJ-167-mer and RJ-167-mer complementary were annealed at a 1:1 molar ratio to RJ-PHIX-42–1, respectively. To generate the 5'Tail DNA substrate, RJ-5'Tail-167mer was annealed at a 1:1 molar ratio to RJ-PHIX-42–1.

2XMBP-BRCA2 wild-type and T1980I proteins were incubated at the indicated concentrations with 400 pM of the radiolabeled DNA substrate for 30 min at 37 °C. The reactions were resolved by electrophoresis on a 6% polyacrylamide gel in 1 X TAE (40 mM Tris–acetate [pH 7.5], 0.5 mM EDTA) buffer for 90 min at 80 Volts. The gel was then dried onto DE81 paper and exposed to a PhosphorImager screen overnight. The screen was scanned on a Molecular Dynamics Storm 840 PhosphorImager and bands were quantified using ImageLab software. The percentage of protein-DNA complexes was calculated as the free radiolabeled DNA remaining in a given lane relative to the protein-free lane, which defined the value of 0% complex, or 100% free DNA.

## Biotin DNA pulldowns

The 2XMBP-BRC purified proteins (80 nM) were incubated with 0, 50, or 100 nM of purified RAD51 for 10 min at 37 °C in Buffer 'S': 25 mM TrisOAc (pH 7.5), 1 mM MgCl$_2$, 2 mM CaCl$_2$, 0.1 μg/μL BSA, 2 mM ATP, and 1 mM DTT. Then 1 nM of the biotinylated ssDNA was added for 10 min at 37 °C. The ssDNA oligonucleotide substrate, RJ-167-mer, was synthesized with a 5' biotin modification and PAGE purified by IDT (Ultramer). Final volumes were normalized with storage buffer as needed. The DNA-protein complexes were then captured by adding 2.5 μL of pre-washed MagnaLink Streptavidin magnetic beads (SoluLink) in buffer SW (Buffer S supplemented with 0.1% Igepal CA-630 and 0.5% Triton X-100). The bead-DNA-protein complexes were rotated at 25 °C for 5 min and then washed 3 X with buffer SW (lacking 2 mM ATP), resuspended in 20 μL laemmli sample buffer, heated at 54 °C for 4 min, and loaded onto a 4–15% gradient SDS-PAGE gel. The amount of RAD51 protein bound and eluted from the biotinylated DNA was determined by western blot using a monoclonal antibody specific to human RAD51 (14B4, Novus).

## Single molecule Förster resonance energy transfer (smFRET)

All reactions were performed at room temperatures in imaging buffer consisting of 50 mM Tris-HCl, pH 8.0, 1 mM MgCl$_2$, 2 mM CaCl$_2$, 2 mM ATP, 0.1 mg/ml BSA, 25% glucose, Trolox and 1% Gloxy (80% of imaging buffer, 20% catalase and 10 mg glucose oxidase) as described in *Marsden et al., 2016*. Twenty nM BRC peptides were pre-incubated with 20 nM RAD51 in a 1:1 ratio for 15 min at 37 °C followed by the addition of the protein complexes to 300 pM DNA that was tethered to a PEG-coated quartz surface through biotin-neutravidin linkage. smFRET assays were performed as described in *Marsden et al., 2016*; *Rothenberg and Ha, 2010*. Briefly, a custom-built optical imaging platform in reference to Olympus IX70 inverted microscope was used, which was coupled to a 532 (10 mWatts) and a 640 (10 mWatts) nm solid -state lasers to excite the sample in the TIRF mode. Photons collected from the Cy3 and Cy5 fluorophores were then imaged on to a single EMCCD camera (Andor iXon3) with images acquired at 33 Hz and EM gain set to 300. Data consisting of 500 frames were recorded with each frame having an exposure time of 30ms and analyzed as indicated in Marsden et al. Matlab (version R2020b) was used to view and analyze the FRET trajectories. The FRET histograms were generated using a sample size of at least 250 single molecule trajectories and the representative histograms did not include zero FRET values or photobleached portions of the FRET trajectories. Origin (version 2021) was used to plot the histograms and the FRET trajectories.

## DNA strand exchange (DOI: 10.1007/978-1-0716-0644-5_8)

All DNA substrates were obtained PAGE purified from IDT. The 3' Tail DNA substrate was generated by annealing RJ-167-mer to RJ-PHIX-42–1 at a 1:1 molar ratio. The dsDNA donor was generated

by first radiolabeling RJ-Oligo1 (5'-TAA TAC AAA ATA AGT AAA TGA ATA AAC AGA GAA AAT A AA G-3') with $^{32}$P (T4 Polynucleotide Kinase) on the 5'-end and annealing it to RJ-Oligo2 (5'-CTT T AT TTT CTC TGT TTA TTC ATT TAC TTA TTT TGT ATT A-3') at a 1:1 molar ratio. The assay buffer contained: 25 mM TrisOAc (pH 7.5), 1 mM $MgCl_2$, 2 mM $CaCl_2$, 0.1 µg/µL BSA, 2 mM ATP, and 1 mM DTT. All pre-incubations and reactions were at 37 °C. The DNA substrates and proteins were at the following concentrations unless otherwise indicated in the figure legend: RPA (0.1 µM); RAD51 (0.4 µM); 3' tail (4 nM molecules); and dsDNA (4 nM molecules). The 3' Tail DNA was incubated first with RPA for 5', followed by the addition of BRCA2 WT or T1980I proteins and RAD51, and finally, the radiolabeled donor dsDNA was added for 30 min. Where proteins were omitted, storage buffer was substituted. The reaction was terminated with Proteinase K/0.5% SDS for 10 min. The reactions were loaded on a 6% polyacrylamide gel in TAE buffer and electrophoresis was at 60 V for 80 min. The gel was then dried onto DE81 paper and exposed to a PhosphorImager screen overnight. The percentage of DNA strand exchange product was calculated as labeled product divided by total labeled input DNA in each lane (for further details see *Jensen et al., 2010*; *Lahiri and Jensen, 2021*).

## Antibodies

| Antibody | Company | Catalog Number, RRID |
|---|---|---|
| MBP | NEB | Cat# E8032L |
| BRCA2 | Abcam | Cat# ab27976 |
| Rabbit Anti-BRCA2 Nt | Bethyl Laboratories | Cat# A303-434 |
| PALB2 | Bethyl Laboratories | Cat# A301-246A |
| RAD51 | Proteintech | Cat#14961–1-AP, RRID: AB_2177083 |
| RAD51 | Novus Biologicals | Cat# NB 100–148, RRID: AB_350083 |
| RAD51 | Abcam ab63801 | Cat# ab63801, RRID: AB_1142428 |
| BrdU-Mouse | BD Biosciences | Cat# 247580 |
| BrdU-Rat | Abcam | Cat# ab6326 RRID:AB_305426 |
| m-IgGκ BP-HRP Antibody | Santa Cruz Biotechnology | Cat# sc-516102, RRID:AB_2687626 |
| goat anti-rabbit IgG-HRP antibody | Santa Cruz Biotechnology | Cat# sc-2004, RRID:AB_631746 |
| Goat anti-Rabbit IgG (H+L) Highly Cross-Adsorbed Secondary Antibody, Alexa Fluor 488 | Thermo Fisher Scientific | Cat# A-11034, RRID:AB_2576217 |
| Goat anti-Mouse IgG (H+L) Cross-Adsorbed Secondary Antibody, Alexa Fluor 546 | Thermo Fisher Scientific | Cat# A-11003, RRID:AB_2534071 |
| Goat anti-Rat Cy5 | Abcam | Cat# ab6565 RRID:AB_955063 |
| Sheep anti-mouse Cy3 | Sigma | Cat# C2181 RRID:AB_258785 |

## Acknowledgements

We thank Nancy Sanchez for assistance with cloning the S1221P BRCA2 mutation and Agata Smog-orzewska for the HA-RAD51 and HA-RAD51 T131P constructs. We thank all members of the Jensen Lab for helpful comments and review of this article. Sudipta Lahiri thanks Rothenberg lab members, especially Dr. Carolus Fijen and Dr. Huijun Xue for help with smFRET experiments and data analysis. The graphical abstract and schematics were created with https://biorender.com/ This research was supported by grants from the NIH (RO1 CA215990), Women's Health Research at Yale, and The Gray Foundation to RBJ, and NIH grants R35 GM134947, AI153040, and CA247773 to ER Further support was provided by The V Foundation BRCA Research collaborative grant (to RBJ & ER), Pfizer (ER), and funds from the Chavkin philanthropic contribution to the Perlmutter Cancer Center (ER), and a Lion Heart Pilot Grant to JJ-S.

## Additional information

### Funding

| Funder | Grant reference number | Author |
|---|---|---|
| National Cancer Institute | CA215990 | Ryan B Jensen |
| Women's Health Research at Yale | | Ryan B Jensen |
| The Gray Foundation | | Ryan B Jensen |
| Lion Heart Pilot Grant | | Judit Jimenez-Sainz |
| National Institutes of Health | R35 GM134947 | Eli Rothenberg |
| National Institutes of Health | AI153040 | Eli Rothenberg |
| National Institutes of Health | CA247773 | Eli Rothenberg |
| The V Foundation BRCA Research | | Eli Rothenberg |
| Pfizer | | Eli Rothenberg |
| Chavkin Philanthropic | | Eli Rothenberg |

The funders had no role in study design, data collection and interpretation, or the decision to submit the work for publication.

### Author contributions

Judit Jimenez-Sainz, Conceptualization, Data curation, Formal analysis, Funding acquisition, Validation, Investigation, Visualization, Methodology, Writing – original draft, Project administration, Writing – review and editing; Joshua Mathew, Formal analysis, Investigation, Visualization, Methodology, Writing – original draft, Writing – review and editing; Gemma Moore, Data curation, Formal analysis, Investigation, Methodology, Writing – review and editing; Sudipta Lahiri, Jennifer Garbarino, Data curation, Formal analysis, Investigation, Visualization, Methodology, Writing – review and editing; Joseph P Eder, Investigation, Writing – review and editing; Eli Rothenberg, Data curation, Funding acquisition, Writing – review and editing; Ryan B Jensen, Conceptualization, Formal analysis, Supervision, Funding acquisition, Investigation, Methodology, Writing – original draft, Writing – review and editing

### Author ORCIDs

Judit Jimenez-Sainz ![ORCID] http://orcid.org/0000-0002-1048-8623
Joshua Mathew ![ORCID] http://orcid.org/0000-0001-5743-2670
Gemma Moore ![ORCID] http://orcid.org/0000-0002-2656-0538
Eli Rothenberg ![ORCID] http://orcid.org/0000-0002-1382-1380
Ryan B Jensen ![ORCID] http://orcid.org/0000-0002-9844-0789

### Decision letter and Author response

Decision letter https://doi.org/10.7554/eLife.79183.sa1
Author response https://doi.org/10.7554/eLife.79183.sa2

## Additional files

### Supplementary files
- Transparent reporting form

### Data availability
All data generated or analyzed during this study are included in the manuscript and supporting files.

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

# Appendix 1

## Appendix 1—key resources table

| Reagent type (species) or resource | Designation | Source or reference | Identifiers | Additional information |
|---|---|---|---|---|
| Gene (*Homo sapiens, Drosophila melanogaster, Mus musculus Felis catus, Rattus norvegicus, Strongylocentrotus purpuratus, Monodelphis domestica, Canis lupus familiaris, Xenopus tropicalis*) | BRCA2 | Uniprot | **P51587** *Homo sapiens (Human)*; Q**9W157** *Drosophila melanogaster (Fruit fly)*; **P97929***Mus musculus (Mouse)*; **Q864S8***Felis catus (Cat)*; **O35923***Rattus norvegicus (Rat)*; **A5A3F7** *Strongylocentrotus purpuratus (Purple sea urchin)*; **A4ZZ89***Monodelphis domestica (Gray short-tailed opossum)*; **Q8MKI9***Canis lupus familiaris (Dog) (Canis familiaris)*; **A4ZZ90***Xenopus tropicalis (Western clawed frog) (Silurana tropicalis)*. | Alignment *Figure 1* |
| Strain, strain background (*Escherichia coli*) | Stellar Competent Cells | Takara-Clontech | Cat# 636766 | Competent cells |
| Strain, strain background (*Escherichia coli*) | Rosetta(DE3)pLysS Competent Cells | Sigma-Aldrich | Cat# 70956 | |
| Strain, strain background (*Escherichia coli*) | *E. coli* Acella(DE3)/ pCH1–RAD51o | **Subramanyam and Spies, 2018** | | |
| Cell line (*Homo-sapiens*) | DLD1 | Horizon Discovery **Hucl et al., 2008** | | |
| Cell line (*Homo-sapiens*) | DLD1 BRCA2 -/- | Horizon Discovery **Hucl et al., 2008** | | |
| Cell line (*Homo-sapiens*) | HEK2093T | **Jensen et al., 2010** | | |
| Recombinant DNA reagent | pBluescript BRCA2 (1–5286 bp) | This paper | | Dr. Ryan Jensen laboratory. Material & Methods section |
| Recombinant DNA reagent | pUC57 BRCA2 (2141–9117 bp) Genescript | Synthetized by Genescript | | |
| Recombinant DNA reagent | pX330-LMNA-gRNA1 | addgene | 122507 | sgRNA plasmid targeting Lamin A |
| Recombinant DNA reagent | pCR2.1 Clover-LMNA Donor | Addgene | 122507 | Donor |
| Transfected construct (*Homo-sapiens*) | 2XMBP-BRC2 | This paper | YLTDENEVGFRGFYSA HGTKLNV STEALQKAVKLFSDIENISE ETSAEVHPISL* | Dr. Ryan Jensen laboratory. Material & Methods section |
| Transfected construct (*Homo-sapiens*) | 2XMBP-BRC2-S1221P | This paper | YLTDENEVGFRGFYPA HGTKLN VSTEALQKAVKLFSDIENISE ETSAEVHPISL* | Dr. Ryan Jensen laboratory. Material & Methods section |
| Transfected construct (*Homo-sapiens*) | 2XMBP-BRC7 | This paper | GKLHKSVSSANTCGIF STASGKS VQVSDASLQNARQVFS EIEDSTKQ* | Dr. Ryan Jensen laboratory. Material & Methods section |
| Transfected construct (*Homo-sapiens*) | 2XMBP-BRC7-T1980I | This paper | GKLHKSVSSANTCGIF SIASGKSV QVSDASLQNARQVFSE IEDSTKQ* | Dr. Ryan Jensen laboratory. Material & Methods section |
| Transfected construct (*Homo-sapiens*) | 2XMBP BRCA2 WT | **Jensen et al., 2010** | | Dr. Ryan Jensen laboratory. Material & Methods section |

*Appendix 1 Continued on next page*

*Appendix 1 Continued*

| Reagent type (species) or resource | Designation | Source or reference | Identifiers | Additional information |
|---|---|---|---|---|
| Transfected construct (*Homo-sapiens*) | 2XMBP BRCA2 S1221P | This paper | | Dr. Ryan Jensen laboratory. Material & Methods section |
| Transfected construct (*Homo-sapiens*) | 2XMBP BRCA2 T1346I | This paper | | Dr. Ryan Jensen laboratory. Material & Methods section |
| Transfected construct (*Homo-sapiens*) | 2XMBP BRCA2 T1980I | This paper | | Dr. Ryan Jensen laboratory. Material & Methods section |
| Peptide, recombinant protein | RPA | This paper, **Anand et al., 2018**; | | Dr. Ryan Jensen laboratory. Material & Methods section |
| Peptide, recombinant protein | RAD51 | This paper, **Subramanyam and Spies, 2018** | | Dr. Ryan Jensen laboratory. Material & Methods section |
| Peptide, recombinant protein | BRC2 peptide | Pierce, This paper | NEVGFRGFYSAHGTKLN VSTEALQKAVKLFSDIEN | Dr. Ryan Jensen laboratory. Material & Methods section |
| Peptide, recombinant protein | BRC2-S1221P peptide | Pierce, This paper | NEVGFRGFYPAHGTKL NVSTEALQKAVKLFSDIEN | Dr. Ryan Jensen laboratory. Material & Methods section |
| Peptide, recombinant protein | BRC7- peptide | Pierce, This paper | SANTCGIFSTASGKSVQ VSDASLQNARQVFSEIED | Dr. Ryan Jensen laboratory. Material & Methods section |
| Peptide, recombinant protein | BRC7- T1980I peptide | Pierce, This paper | SANTCGIFSIASGKSVQV SDASLQNARQVFSEIED | Dr. Ryan Jensen laboratory. Material & Methods section |
| Antibody | Anti-MBP (mouse monoclonal) | NEB | Cat# E8032L | IF (1:100) WB (1:1000) |
| Antibody | Anti-BRCA2 (rabbit polyclonal) | Abcam | Cat# ab27976 | IF (1:100) WB (1:1000) |
| Antibody | Anti-BRCA2 Nt (rabbit polyclonal) | Bethyl Laboratories | Cat# A303-434 | WB (1:1000) |
| Antibody | Anti-RAD51 (rabbit polyclonal) | Proteintech | Cat# 14961–1-AP, RRID:AB_2177083 | IF (1:100) WB (1:1000) |
| Antibody | Anti-RAD51 (mouse monoclonal) | Novus Biologicals | Cat# NB 100–148, RRID:AB_350083 | WB (1:1000) |
| Antibody | Rabbit Anti-RAD51 (rabbit polyclonal) | Abcam | Cat# ab63801, RRID:AB_1142428 | IF (1:100) |
| Antibody | Anti-BrdU (purified mouse monoclonal) | BD Biosciences | Cat# 347580, RRID:AB_10015219 | 1:10 |
| Antibody | Anti-BrdU (rat monoclonal) | Abcam | Cat# ab6326, RRID:AB_2313786 | 1:40 |
| Antibody | Anti-Rat Cy5 (goat polyclonal) | Abcam | Cat# ab6565, RRID:AB_955063 | 1:100 |
| Antibody | Anti-Mouse Cy3 (sheep polyclonal) | Sigma Aldrich | Cat# C2181, RRID:CVCL_K942 | 1:100 |
| Antibody | m-IgGκ BP-HRP | Santa Cruz Biotechnology | Cat# sc-516102, RRID:AB_2687626 | WB (1:10000) |
| Antibody | anti-rabbit IgG-HRP (goat polyclonal) | Santa Cruz Biotechnology | Cat# sc-2004, RRID:AB_631746 | WB (1:10000) |
| Antibody | anti-Rabbit IgG (H+L) Highly Cross-Adsorbed Secondary Antibody, Alexa Fluor 488 (goat polyclonal) | Thermo Fisher Scientific | Cat# A-11034, RRID:AB_2576217 | IF (1:10000) |

*Appendix 1 Continued on next page*

*Appendix 1 Continued*

| Reagent type (species) or resource | Designation | Source or reference | Identifiers | Additional information |
|---|---|---|---|---|
| Antibody | anti-Mouse IgG (H+L) Cross-Adsorbed Secondary Antibody, Alexa Fluor 546 (goat polyclonal) | Thermo Fisher Scientific | Cat# A-11003, RRID:AB_2534071 | IF (1:10000) |
| Sequenced-based reagent | dT40 | *Carreira and Kowalczykowski, 2011* | EMSA | 5'TTTTTTTTTTTTTTTTTTTTTTTTTTTTTTTTTTTTTTTT3' |
| Sequenced-based reagent | RJ-167-mer | *Jensen et al., 2010* | EMSA DNA strand exchange | 5'-CTG CTT TAT CAA GAT AAT TTT TCG ACT CAT CAG AAA TAT CCG TTT CCT ATA TTT ATT CCT ATT ATG TTT TAT TCA TTT ACT TAT TCT TTA TGT TCA TTT TTT ATA TCC TTT ACT TTA TTT TCT CTG TTT ATT CAT TTA CTT ATT TTG TAT TA TCC TTA TCT TAT TTA-3' |
| Sequenced-based reagent | RJ-5'tail 167-mer | *Jensen et al., 2010* | EMSA | : 5'-ATT TAT TCT ATT CCT CTT TAT TTT CTC TGT TTA TTC ATT TAC TTA TTT TGT ATT AAT TTC CTA TAT TTT TTA CTT GTA TTT CTT ATT CAT TTA CTT ATT TTG TAT TAT CCT TAT TTA TAT CCT TTC TGC TTT ATC AAG ATA ATT TTT CGA CTC ATC AGA AAT ATC CG-3' |
| Sequenced-based reagent | RJ-PhiX-42–1 | *Jensen et al., 2010* | EMSA DNA Strand Exchange | 5'-CGG ATA TTT CTG ATG AGT CGA AAA ATT ATC TTG ATA AAG CAG-3' |
| Sequenced-based reagent | PS4 | *Jensen et al., 2010* | EMSA DNA Strand Exchange | 5'- TAATACAAAATAAGTA AATGA ATAAACAGAGAAAATAAAG –3' |
| Sequenced-based reagent | PS5 | *Jensen et al., 2010* | EMSA DNA Strand Exchange | 5'- CTTTATTTTCTCTGTT TATTCATT TACTTATTTTGTATTA –3' |
| Chemical compound, drug | Jet Optimus | Polyplus Transfection | 117–07 | |
| Chemical compound, drug | Calcium Phosphate transfection | This paper | https://www.nature.com/articles/nmeth0405-319 | Dr. Ryan Jensen laboratory. Material & Methods section |
| Chemical compound, drug | Lipofectamine3000 | Thermo Scientific | L3000001 | |
| Chemical compound, drug | TurboFect reagent | Thermo Scientific | R0533 | |
| Chemical compound, drug | G418 | American Bio | AB05058-00020 | |
| Chemical compound, drug | protease inhibitor cocktail | Sigma Aldrich | 04693159001 | |
| Chemical compound, drug | amylose resin | NEB | E8021 | |
| Chemical compound, drug | Hydroxyurea | Sigma Aldrich | 127-07-1 | |
| Chemical compound, drug | Benzonase | Millipore | 9025-65-4 | |

*Appendix 1 Continued on next page*

*Appendix 1 Continued*

| Reagent type (species) or resource | Designation | Source or reference | Identifiers | Additional information |
|---|---|---|---|---|
| Chemical compound, drug | MMC | Sigma Aldrich | M4287 | |
| Chemical compound, drug | Cisplatin | Sigma Aldrich | 15663-27-1 | |
| Chemical compound, drug | Olaparib | ApexBio | AZD2281, Ku-0059436 | |
| Chemical compound, drug | Talazoparib-BMN673 | ApexBio | A4153 | |
| Chemical compound, drug | DTT | Sigma Aldrich | 3483-12-3 | |
| Chemical compound, drug | IdU | Sigma Aldrich | I7125 | |
| Chemical compound, drug | CldU | Sigma Aldrich | C6891 | |
| Chemical compound, drug | FluorSave reagent | Calbiochem | 345789 | |
| Software, algorithm | ClustalX | *Larkin et al., 2007* | http://www.ch.embnet.org/software/ClustalW.html | |
| Software, algorithm | Bioedit | Bioedit (*Hall, 1999*) | https://itservices.cas.unt.edu/software/bioedit725 | |
| Software, algorithm | Snapgene 5.3.1. | Snapgene 5.3.1. | from Insightful Science; available at snapgene.com | |
| Software, algorithm | Prism Graph Pad version 9.0 | https://www.graphpad.com/updates/prism-900-release-notes | | |
| Software, algorithm | PyMOL Molecular Graphics System, Version 1.3 Schrödinger, LLC. | Schrodinger, L. (2010) The PyMOL Molecular Graphics System | | |
| Software, algorithm | Image J | *Schneider et al., 2012* | https://imagej.nih.gov/ij/ | |
| Software, algorithm | ImageQuant TL 8.0 image analysis software | http://www.hhmi.umbc.edu/downloads/Imaging%20support%20GE/IQ%20TL%20collateral/IQTL_UserManual%208.pdf | | |
| Software, algorithm | Image Lab Software 6.1 (Bio-Rad) | https://www.bio-rad.com/en-us/product/image-lab-software?ID=KRE6P5E8Z&WT.mc_id=220411034041&WT.knsh_id=_kenshoo_clickid_&gclid=Cj0KCQjwr4eYBhDrARIsANPywCiJded2fEKfZWaf_-Fxki9ynO1VFgLb3kkN-gJ-A3u4baPwfvKj2ZQaApk-EALw_wcB | | |
| Software, algorithm | ChemiDoc MP imaging system XRS+ (Bio-Rad) | https://www.bio-rad.com/en-us/product/chemidoc-mp-imaging-system?ID=NINJ8ZE8Z&gclid=Cj0KCQjwr4eYBhDrARIsANPywCjmMNj5mMKTHZkXXj4iOrhnRLMo8JRt4nwLib3XTupvbHi2Q2CRJnkaAvp6EALw_wcB | | |
| Software, algorithm | Cell profiler software following counting and scoring pipeline | *Carpenter et al., 2006* | | |
| Other | PDB structure BRC4 | *Pellegrini et al., 2002* | 1NOW | Swiss model BRC2-7 |
| Other | DAPI stain | Thermo Fisher | D1306 | (1 µg/mL) |
| Other | NucRed Live 647 ReadyProbes Reagent | Thermo Fisher | R37106 | (1 µg/mL) |

