## [Editor Report]

This study provides a thorough functional analysis of three mutations in the BRCA2 gene that do not seem to necessarily cause breast cancer. The authors use functional assays in cancer cells and with recombinant proteins to determine that two BRCA2 variants, S1221P and T1980I, are indeed pathogenic, while the T13461 variant is fully functional and benign. The strength of the study is the rigorous assessment of these mutations in a variety of established assays for BRCA2, and has improved significantly following the review process. The work should have a broad impact in the breast cancer field.

---

## [Decision Letter]

**Decision letter after peer review:**

Thank you for submitting your article "BRCA2 BRC missense variants disrupt RAD51-dependent DNA repair" for consideration by *eLife*. Your article has been reviewed by 3 peer reviewers, and the evaluation has been overseen by a Reviewing Editor and Päivi Ojala as the Senior Editor. The following individual involved in the review of your submission has agreed to reveal their identity: Sarah R Hengel (Reviewer #2).

Essential revisions:

1) Figure 1 It would be helpful to add a control with DLD1 BRCA2-proficient cells for visualization of the endogenous BRCA2 expression level.

2) Figure S3. Quantitation of the plating efficiencies needs to be included.

3) Figure 5B shows a disproportionally stronger decrease in the free DNA bands than an increase in the RAD51-ssDNA complexes, which may indicate the presence of nucleases. Were the nucleases tested in the protein preps?

4) Figure 5E. In fact, BRC7-T1980I shows a noticeable increase in RAD51 capture.

5) Abstract: It may be helpful to tune readers into what kind of cells you are performing your analysis with: DLD-1 colorectal cancer cells – think about adding it to the abstract.

6) Page 3: When introducing DBD of BRCA2, the NLS info should be mentioned here in coordination with DSS1. As written the NLS is described only in context to the C-terminus.

7) Page 4: What is the total number of missense mutations in the whole BRCA2 protein? And how does this compare to the number of missense mutations in the BRC repeats? While the authors note that VUSs are frequently found to be unique to individual families – are there missense mutations that are more frequently populated? How did you focus on the two mutations within BRC repeats – was it first using a reductionist approach to focus in on amino acid conservation?

8) Page 15: It is unclear if the BRCA2 T1346I variant has been published before? Is there anything else known about this mutation and the tumor mutational burden? What happened to the patient? Did they survive?

9) Page 16: Explicitly state which modeling algorithms were used and how the computational structural analysis was performed in the text of the manuscript.

10) Page 16: Adjust this sentence and describe the loop either in amino acids etc… (However, polar contacts within a loop," is a bit vague and I am not certain what the authors are trying to explain.

11) Page 17: "….re-introduction of recombinant BRCA2 protein in deficient cells increases total expression of RAD51" Do you think this is a chromatin or protein phenomenon. It would be interesting to hear about the authors' thoughts in the discussion on this repeatable result.

12) When describing cell-based assays throughout the manuscript text please explicitly list the cell type-either DLD-1 or HEK293 cells within the manuscript where the assays are described (Example-page 18 affects HDR (Figure 3A).

13) Page 19: Title: Individual S1221P and T1980I BRC mutations eliminate binding to RAD51 I suggest adding DLD-1 cells at the end of the title. Would the same conclusions be drawn if the assays were performed in other cell lines? It may be useful to be more specific in language.

14) Page 20: It has been shown in our field using single-molecule FRET assays that measure RAD51 nucleation, that RAD51 nucleates as dimers and multimers of dimers etc via transition density hidden Markov analysis (Subramanyam et al., 2018 (PMID: 29458759)). This paper should be cited in the sentence that begins with "To reconcile our pulldown-data….."

15) Page 22: Could it be possible that BRC7 may be changing the conformation of RAD51 to adopt a more ATP-bound or open conformation responsible for the lower FRET state (0.7)? FRET values do not allow the authors to conclude that RAD51 nucleation is stimulated. These data simply show a change in the conformation of the DNA substrate. I would suggest re-wording this section. It is also plausible that BRC2WT would show a lower FRET value at higher protein concentrations than those assayed (20 nM BRC2WT + 20 nM RAD51). Did the authors try increasing the protein concentrations in this assay?

16) Page 22: What is the photobleaching lifetime in the FRET experiments? This should be included for publication as well as the buffer components and any photostabilizers utilized. The imaging time etc should also be reported. Were there any lifetime differences between the various mutants?

17) It is conceivable that the impact of the study could be improved with the presentation of the data, particularly in the Discussion. Namely – what new information do we have about the BRC repeat function from this study? Alternatively, perhaps the authors could explain why they chose to test 3 alleles in the BRC repeat region in lots of assays, instead of large-scale testing of VUSs in the BRC repeat region in a couple of assays?

---

## [Author Response]

Essential revisions:1) Figure 1 It would be helpful to add a control with DLD1 BRCA2-proficient cells for visualization of the endogenous BRCA2 expression level.

We believe the reviewer is referring to the western blot in Figure 2A. We have performed the requested experiment which is included in the manuscript as Figure 2-supplement figure 1 revised. As well, we have included the corresponding changes in the figure legend and manuscript file.

2) Figure S3. Quantitation of the plating efficiencies needs to be included.

We believe the reviewer is referring to Figure 2-supplement figure 2. We have quantitated the plating efficiencies as requested now included as figure2-supplement figure 2B and figure2-supplement figure 2-source data 1-revised plating efficiency.

3) Figure 5B shows a disproportionally stronger decrease in the free DNA bands than an increase in the RAD51-ssDNA complexes, which may indicate the presence of nucleases. Were the nucleases tested in the protein preps?

Yes, all purified proteins were found to be free of contaminating nucleases. Please see included file depicting nuclease assays performed on purified proteins.

4) Figure 5E. In fact, BRC7-T1980I shows a noticeable increase in RAD51 capture.

We do note that BRC7-T1980I shows a 2.5-fold reduction in RAD51 capture compared to WT BRC7 and yet exhibits higher levels of RAD51 bound than the RAD51 alone (lane 3). Perhaps there is a residual binding mode specific to the RAD51 filament that is retained by the BRC7-T1980I mutant, however, extensive further characterization of BRC-RAD51 binding modes will be required to understand the specific interactions which is beyond the scope of this manuscript. We now include discussion of this result in the manuscript (page 24).

“Interestingly, residual RAD51-ssDNA binding was observed with BRC7-T1980I (compare lane 15 to lane 3 in Figure 5E) perhaps indicative of a binding mode specific to BRC7 that is not completely diminished by the T1980I mutation.”

5) Abstract: It may be helpful to tune readers into what kind of cells you are performing your analysis with: DLD-1 colorectal cancer cells – think about adding it to the abstract.

We appreciate the reviewer’s suggestion; however, we have done extensive analysis in both DLD1 and HEK293T cells. To help guide readers, we now explicitly state which cell line was used in each experiment throughout the manuscript.

6) Page 3: When introducing DBD of BRCA2, the NLS info should be mentioned here in coordination with DSS1. As written the NLS is described only in context to the C-terminus.

To clarify this point, we have added some text on pages 3 and 4:

“A nuclear export sequence (NES) in the DBD has been found to overlap with the binding region for DSS1 and a specific missense mutation (D2723H) that disrupts DSS1 binding results in BRCA2 export to the cytoplasm (Jeyasekharan et al., 2013).”

“Putative nuclear localization signals (NLSs) are located in the CTD of BRCA2, however, surprisingly, it remains unclear how exactly nuclear/cytoplasmic trafficking of BRCA2 is regulated (Bertwistle et al., 1997; Han et al., 2008; Spain et al., 1999; Yano et al., 2000) (reviewed in Jimenez-Sainz and Jensen, 2021).

“We and others have recently discovered that missense mutations in the DBD mislocalized BRCA2 to the cytosol leading to HR deficiency and sensitivity to crosslinking agents and PARPi (Jeyasekharan et al., 2013; Lee et al. 2021; Jimenez-Sainz and Jensen, 2021; Jimenez-Sainz, 2022).”

7) Page 4: What is the total number of missense mutations in the whole BRCA2 protein? And how does this compare to the number of missense mutations in the BRC repeats? While the authors note that VUSs are frequently found to be unique to individual families – are there missense mutations that are more frequently populated? How did you focus on the two mutations within BRC repeats – was it first using a reductionist approach to focus in on amino acid conservation?

The total number of BRCA2 variants in Clin Var (accessed 2022-07-12) is 15,079 of which 6,541 are missense. The total number of BRC variants is 4,234 of which 2,114 are missense. Thus, almost 50% of the BRC variants are missense and 32% of missense variants are located in the BRC domain. Most missense variants in the BRC domain are of uncertain significance (97%) therefore it is unknown how mutations in this essential domain could affect cancer risk.

Clinically, most germline truncating pathogenic mutations in BRCA2 occur in the BRC repeat region. Because secondary reversion events that lead to in-frame deletions cluster in the BRC repeats, we feel it is important to fully characterize this region to further understand both chemotherapeutic resistance mechanisms as well as the biological role of the BRC repeats. Additionally, germline or somatic missense mutations in the BRC repeats are not well characterized and likely contribute to cancer risk and therapy response.

In the breast cancer information core database (BIC), there are three common missense mutations in the BRC repeats: D1420Y (Submissions 200, Exact 0.0068, https://www.ncbi.nlm.nih.gov/clinvar/variation/41549/), R2034C (Submissions 104, Exact 0.0032, https://www.ncbi.nlm.nih.gov/clinvar/variation/41558/?new_evidence=false) and D1902N (Submissions 87, Exact 0.0017, https://www.ncbi.nlm.nih.gov/clinvar/variation/51912/?new_evidence=false).

However, these missense mutations have been reviewed by an expert panel designated as benign.

Based on the analysis presented in this study, we predict that D1420Y, R2034C, and D1902N do not alter RAD51 binding and/or function due to their locations in the spacer regions between BRC repeats as well as lack of conservation across species. (D1420Y is in the spacer between BRC2 and BRC3, D1902N is in the spacer between BRC6 and BRC7, and R2034C is in the spacer between BRC7 and BRC8).

We initially focused on the T1980I variant as its location within the conserved FxTAS motif predicted disruption of RAD51 binding. Previously, we and others had shown that BRC repeats 1-4 behave differently than BRC repeats 5-8 (Carreira and Kowalczykowski, 2011; Chatterjee et al., 2016) and given that T1980I resides in BRC7, we decided to characterize S1221P in BRC2 to compare BRC repeats from the two modules. Again, due to the location of the S1221P mutation in the FxTAS motif, it seemed likely that this variant would disrupt RAD51 binding. The T1346I variant was brought to our attention by Paul Eder, MD, a medical oncologist at Yale, who was treating a colon cancer patient and enrolling patients into a PARP inhibitor clinical trial. The variant was identified by whole exome sequencing of the patient’s colon tumor. We found T1346I of interest as the mutation was located in the spacer region between BRC repeats and it was unclear if this variant was pathogenic or benign.

8) Page 15: It is unclear if the BRCA2 T1346I variant has been published before? Is there anything else known about this mutation and the tumor mutational burden? What happened to the patient? Did they survive?

This is the first report of the T1346I variant to our knowledge. Three other missense changes in the T1346 residue have been deposited in ClinVar as variants of uncertain significance (T1346A, T1346S and T1346N) (https://www.ncbi.nlm.nih.gov/clinvar/?term=brca2%5Bgene%5D+T1346).

The patient did not have any known family history of cancer and did not respond to standard of care chemotherapy (FOLFOXIRI) [ 5 – Fluorouracil/leucovorin/oxaliplatin/irinotecan]. The patient declined to enroll in clinical trial NCT02576444 (olaparib combinations), elected hospice care, and died soon thereafter.

9) Page 16: Explicitly state which modeling algorithms were used and how the computational structural analysis was performed in the text of the manuscript.

The requested information is now included on page 6 (Material and Methods section) and on page 17 (Results section).

“SWISS-Model website: "Models are computed by the SWISS-MODEL server homology modelling pipeline (Waterhouse et al.) which relies on ProMod3 (Studer et al.), an in-house comparative modelling engine based on OpenStructure (Biasini et al.).”

10) Page 16: Adjust this sentence and describe the loop either in amino acids etc… (However, polar contacts within a loop," is a bit vague and I am not certain what the authors are trying to explain.

The sentence has been changed to clarify:

“Modeling of the T1526A substitution does not lead to steric clashing, however, could disassemble the BRC4 loop conformation (hydrophobic contacts F1524, A1527 and K1530 and polar contacts T1526 and S1528) necessary for RAD51 binding (Figure 1—figure supplement 1) (Carreira *et al.*, 2009). ”

11) Page 17: "….re-introduction of recombinant BRCA2 protein in deficient cells increases total expression of RAD51" Do you think this is a chromatin or protein phenomenon. It would be interesting to hear about the authors' thoughts in the discussion on this repeatable result.

We think BRCA2 and RAD51 protein levels are mutually dependent as described in this publication (https://pubmed.ncbi.nlm.nih.gov/24210700/) and we speculate that BRCA2 and RAD51 positively regulate the stability of each other. In supplementary figure 6 of our previous publication (Chatterjee et al. 2016, NAR) re-introduction of BRC1-8 or full-length BRCA2 increased the total expression of RAD51 in cellular assays and this observation was discussed in detail in the Discussion section of the prior publication. Further analysis will be required to define the molecular regulation of protein stability between the BRCA2 and RAD51 proteins.

12) When describing cell-based assays throughout the manuscript text please explicitly list the cell type-either DLD-1 or HEK293 cells within the manuscript where the assays are described (Example-page 18 affects HDR (Figure 3A).

As requested, the manuscript was revised to specify whether DLD1 or 293T cells were utilized on pages 19, 20 and 21

13) Page 19: Title: Individual S1221P and T1980I BRC mutations eliminate binding to RAD51 I suggest adding DLD-1 cells at the end of the title. Would the same conclusions be drawn if the assays were performed in other cell lines? It may be useful to be more specific in language.

Indeed, experiments were performed in DLD-1 and 293T cells as well as using purified proteins all coming to the same conclusion. The manuscript was updated to specify which cell lines were used or whether purified proteins were used.

14) Page 20: It has been shown in our field using single-molecule FRET assays that measure RAD51 nucleation, that RAD51 nucleates as dimers and multimers of dimers etc via transition density hidden Markov analysis (Subramanyam et al., 2018 (PMID: 29458759)). This paper should be cited in the sentence that begins with "To reconcile our pulldown-data….."

We thank the reviewer for bringing this to our attention. We have now added the citation.

15) Page 22: Could it be possible that BRC7 may be changing the conformation of RAD51 to adopt a more ATP-bound or open conformation responsible for the lower FRET state (0.7)? FRET values do not allow the authors to conclude that RAD51 nucleation is stimulated. These data simply show a change in the conformation of the DNA substrate. I would suggest re-wording this section. It is also plausible that BRC2WT would show a lower FRET value at higher protein concentrations than those assayed (20 nM BRC2WT + 20 nM RAD51). Did the authors try increasing the protein concentrations in this assay?

While we acknowledge that smFRET can not directly measure RAD51 nucleation/filament formation, given the extensive literature on the ability of RAD51 to nucleate, form filaments, and stretch DNA, we argue that this is the behavior we are observing. Nonetheless, we appreciate the reviewer’s comment regarding the idea that BRC repeats may induce conformational changes in RAD51 translating into conformational changes in the DNA substrate and will include this possible interpretation in the Discussion section of the manuscript.

An interesting point concerns the way BRC2 and BRC7 may bind RAD51 differently. If BRC2 and BRC7 had equivalent binding modalities to RAD51, one would expect that both BRCs (2 and 7) would contribute to a similar extent in RAD51 conformational dynamics. But we do not see that in our experiments. As the reviewer rightfully pointed out, our measurements only show changes in DNA dynamics, therefore, we concluded that since RAD51 and BRC7 individually (at 20 nM) did not show any change in DNA conformation, but together they did, and given that BRC7 has a preference for RAD51 filaments, it is possible that BRC7 is stimulating RAD51 nucleation and/or filament formation in a different manner than BRC2.

As mentioned in the results, we performed experiments with BRC2 and BRC7 in the concentration range of 50-100 nM. We note that even at higher concentrations of BRC7, we observed the same magnitude of FRET change with 20 nM RAD51 and observed no change with BRC2. We could not perform the experiments with higher concentrations of RAD51, as even with 40-50 nM RAD51, we see low FRET populations consistent with what was reported previously by Maria Spies and co-workers (Subramanyam et al., 2018 (PMID: 29458759)). Under our conditions, 20 nM RAD51 was the highest concentration obtainable that did not result in a change in FRET but allowed us to measure stimulation by BRC7.

16) Page 22: What is the photobleaching lifetime in the FRET experiments? This should be included for publication as well as the buffer components and any photostabilizers utilized. The imaging time etc should also be reported. Were there any lifetime differences between the various mutants?

We apologize for the lack of clarity but photobleached samples were not included in our analysis. Moreover, we are using a method that was previously published (citation number 35 (page 32)):

“Marsden, C.G., Jensen, R.B., Zagelbaum, J., Rothenberg, E., Morrical, S.W., Wallace, S.S., and Sweasy, J.B. (2016). The Tumor-Associated Variant RAD51 G151D Induces a Hyper-Recombination Phenotype. PLoS Genet *12*, e1006208. 10.1371/journal.pgen.1006208.”.

However, as per reviewer’s suggestion, we have updated the methods and provided additional details.

Single molecule Förster Resonance Energy Transfer (smFRET)

“All reactions were performed at room temperatures in imaging buffer consisting of 50 mM Tris-HCl, pH 8.0, 1 mM MgCl2, 2 mM CaCl_2_, 2 mM ATP, 0.1 mg/ml BSA, 25% glucose, Trolox and 1% Gloxy (80% of imaging buffer, 20% catalase and 10 mg glucose oxidase) as described in Marsden et al. (Marsden et al., 2016). 20 nM BRC peptides were pre-incubated with 20 nM RAD51 in a 1:1 ratio for 15 minutes at 37 ^o^C followed by the addition of the protein complexes to 300 pM DNA that was tethered to a PEG-coated quartz surface through biotin-neutravidin linkage. smFRET assays were performed as described in (Marsden et al., 2016; Rothenberg and Ha, 2010). Briefly, a custom-built optical imaging platform in reference to Olympus IX70 inverted microscope was used, which was coupled to a 532 (10 mWatts) and a 640 (10 mWatts) nm solid -state lasers to excite the sample in the TIRF mode. Photons collected from the Cy3 and Cy5 fluorophores were then imaged on to a single EMCCD camera (Andor iXon3) with images acquired at 33 Hz and EM gain set to 300. Data consisting of 500 frames were recorded with each frame having an exposure time of 30 ms and analyzed as indicated in Marsden et al. Matlab (version R2020b) was used to view and analyze the FRET trajectories. The FRET histograms were generated using a sample size of at least 250 single molecule trajectories and the representative histograms did not include zero FRET values or photobleached portions of the FRET trajectories. Origin (version 2021) was used to plot the histograms and the FRET trajectories.”

17) It is conceivable that the impact of the study could be improved with the presentation of the data, particularly in the Discussion. Namely – what new information do we have about the BRC repeat function from this study? Alternatively, perhaps the authors could explain why they chose to test 3 alleles in the BRC repeat region in lots of assays, instead of large-scale testing of VUSs in the BRC repeat region in a couple of assays?

The goal of this study was two-fold: (1) to identify novel pathogenic variants within the BRC repeats of BRCA2 with implications for cancer risk in patients, and (2) to leverage deleterious BRC variants to elucidate the specific functions of each BRC repeat within the context of the full-length BRCA2 protein. Our study revealed several novel insights including: (1) the findings that individual BRC repeat mutations impair the HDR functions within the context of the full-length BRCA2 protein in both cellular and biochemical assays despite the presence of the remaining functional BRC repeats, (2) BRC7 alone can bind RAD51, (3) a tumor-derived variant located within a spacer region between BRC repeats did not alter BRCA2 function, (4) BRC7 appears to stimulate RAD51 nucleation/filament stability in a different manner than BRC2, and (5) utilizing the RAD51 T131P mutation (incapable of self-association and filament formation), we confirmed binding to BRC2 but not BRC7, providing additional evidence that BRC repeats 1-4 engage monomeric RAD51 whereas BRC repeats 5-8 interact with the filament form of RAD51.

An important point regarding large-scale studies of BRCA2 VUS is that the specific functions of BRCA2 required for tumor suppression have not been rigorously identified. Our assumption that homology-directed repair (RAD51 loading and stimulation of RAD51 nucleation) is the key mechanism seems likely but still needs to be proven. BRCA2 is a large, complex protein with both caretaker and tumor suppressor functions that are paradoxically required for viability, and yet, loss of BRCA2 function in somatic cells drives tumor initiation and/or progression. In-depth analysis of the underlying biological role of specific BRCA2 domains, including the modular BRC repeats, will help elucidate the features required for both cell viability and tumor suppressor functions. For large-scale testing of BRCA2 VUS to be successful to predict cancer risk, we need to carefully consider the specific functions analyzed in the assay (rescue of viability? canonical HDR functions? fork protection?). To date, there is no assay that directly measures the tumorigenic capacity of BRCA2 mutations as has been done for certain oncogenes. With this in mind, we should proceed with caution in terms of clinical integration of functional assays into genetic counseling until we fully understand how specific variants will impact the tumor suppressor functions of BRCA2.